# From the Jordan Product to Riemannian Geometries on Classical and Quantum States

**DOI:** 10.3390/e22060637

**Published:** 2020-06-08

**Authors:** Florio M. Ciaglia, Jürgen Jost, Lorenz Schwachhöfer

**Affiliations:** 1Max Planck Institute for Mathematics in the Sciences, 04103 Leipzig, Germany; jjost@mis.mpg.de; 2Faculty for Mathematics, TU Dortmund University, 44221 Dortmund, Germany; lschwach@math.tu-dortund.de

**Keywords:** information geometry, Fisher–Rao metric tensor, quantum states, Fubini–Study metric tensor, Bures–Helstrom metric tensor, Jordan product, differential geometry of C*-algebras

## Abstract

The Jordan product on the self-adjoint part of a finite-dimensional $C^{*}$-algebra A is shown to give rise to Riemannian metric tensors on suitable manifolds of states on A, and the covariant derivative, the geodesics, the Riemann tensor, and the sectional curvature of all these metric tensors are explicitly computed. In particular, it is proved that the Fisher–Rao metric tensor is recovered in the Abelian case, that the Fubini–Study metric tensor is recovered when we consider pure states on the algebra B(H) of linear operators on a finite-dimensional Hilbert space H, and that the Bures–Helstrom metric tensors is recovered when we consider faithful states on B(H). Moreover, an alternative derivation of these Riemannian metric tensors in terms of the GNS construction associated to a state is presented. In the case of pure and faithful states on B(H), this alternative geometrical description clarifies the analogy between the Fubini–Study and the Bures–Helstrom metric tensor.

## 1. Introduction

The study of geometrical structures on the space of classical and quantum states is a well-developed and constantly growing subject. In the context of classical probability theory, it is almost impossible to overestimate the impact of the Fisher–Rao metric tensor $G_{FR}$ introduced by Rao in [1] based on the Fisher information matrix introduced in [2] (see also [3,4,5,6,7]). From the theoretical point of view, this metric tensor on the space of probability measures is characterized by a universality property, that is, it is the only Riemannian metric tensor (up to a constant number) that is equivariant with respect to the family of Markov morphisms between probability spaces (see [6,8,9,10,11] for more detailed discussions and for the proofs of this statement in various contexts). In the quantum context, that is, when probability distributions are “replaced” by density operators on the Hilbert space of the system, the situation becomes more involved because, as Petz showed in [12] finishing the work started by Cencov and Morozowa in [13], there is an infinite number of metric tensors on the manifold of (invertible) density operators satisfying the quantum version of the equivariance property of the Fisher–Rao metric tensor, namely, the equivariance with respect to the class of completely-positive, trace-preserving maps. Furthermore, much has been discovered (and is still being discovered), on the link between these metric tensors and families of quantum relative entropies (see [14,15,16,17,18]). Quite interestingly, but not completely surprising, all these metric tensors reduce to the Fisher–Rao metric tensor if a suitable classical limit is performed.

From a purely theoretical point of view, there is no need to restrict our attention to geometrical structures on quantum states associated with ***covariant*** tensor fields as in the case of metric tensors discussed above. Indeed, in [19,20,21,22,23,24,25], the associative product of the algebra B(H) of linear operators on the finite-dimensional Hilbert space H associated with a quantum system has been suitably exploited to define two ***contravariant*** tensor fields on the space of self-adjoint operators on H, and these tensor fields have been used to give a geometrical description of the Gorini–Kossakowski–Lindblad–Sudarshan (GKLS) equation describing the dynamical evolution of open quantum systems (see [19,21,22,24,26,27,28]). These two tensor fields, named Λ and R, are associated with the antisymmetric part (the Lie product) and the symmetric part (the Jordan product) of the associative product in B(H), respectively. In particular, the tensor field Λ turns out to be the Poisson tensor associated with the coadjoint action of the unitary group U(H) on the space of self-adjoint operators on H, when the latter is thought of as the dual space of the Lie algebra u(H). Consequently, it makes sense to study the symplectic foliation associated with Λ, and it turns out that the symplectic leaves of Λ passing through quantum states are the manifolds of isospectral density operators, in particular, the leaf passing through a pure state is diffeomorphic to the complex projective space of the Hilbert space H of the system. From the point of view of open quantum dynamics, the manifolds of isospectral quantum states are not “big enough” in the sense that the dynamical evolution associated with the GKLS equation will cross these manifolds transversally unless it coincides with the unitary evolution of a closed system. Furthermore, from the point of view of quantum information theory, the manifold on which the metric tensors appearing in Petz’s classification are defined is the manifold of invertible quantum states, which is the union of all the manifolds of isospectral quantum states with maximal rank. Consequently, the geometry of Λ is not enough to capture all the relevant features of quantum states.

In [29,30,31], it has been shown that the relevant manifolds of quantum states may be described as submanifolds of homogeneous spaces of the group GL(H) of invertible operators on H. Specifically, if ξ denotes a self-adjoint operator on H, the group GL(H) acts on the space of self-adjoint operators according to the map $\xi \mapsto g\;\xi \;g^{†}$, where † denotes the adjoint operator on B(H). This action does not preserve the spectrum, and in particular the trace, of ξ unless g is unitary, however, it preserves the positivity of ξ in the sense that $g\;\xi \;g^{†}$ is positive semi-definite if ξ is positive semi-definite, and it preserves the rank of ξ. Furthermore, the orbit of GL(H) through a positive semi-definite operator turns out to be a submanifold of the space of self-adjoint operators on H, which is a homogeneous space of GL(H) consisting of all the positive semi-definite operators with the same rank. Then, on each of these orbits, it is possible to select all those elements with unit trace and prove that this set is actually a submanifold of the orbit which, by construction, consists of all the quantum states with fixed rank. In particular, we have two extreme cases: the manifold of pure quantum states consisiting of quantum states with rank equal to 1; the manifold of invertible quantum states consisting of quantum states with maximal rank. The manifold of quantum states with maximal rank thus obtained coincides with the manifold of invertible quantum states on which Petz’s metric tensors are defined. Furthermore, it is possible to link the tensor fields Λ and R with the action of GL(H) by showing that the vector fields associated with linear functions by means of Λ and R provide the representation of the Lie algebra of GL(H) integrating to the action of GL(H) described above (see [20,22]).

In this work, we show that the orbits of GL(H) through positive semi-definite operators behave, with respect to R, as the symplectic leaves (e.g., the manifolds of isospectral states) behave for the Poisson tensor Λ. Specifically, we show that R is invertible on each of these orbits and its inverse gives rise to a Riemannian metric tensor the geometry of which we characterize by computing the covariant derivative, the geodesics, the Riemann tensor, and the sectional curvature. Then, we study the Riemannian geometries on the manifolds of quantum states with the same rank arising from the fact that each of these manifolds is a submanifold of a given orbit of GL(H) through positive semi-definite operators.

To be more precise, in Section 2, we review the geometrical aspects of the space of states of a finite-dimensional $C^{*}$-algebra that we will need in the rest of the work. In Section 3 and Section 4, we actually prove the aforementioned results not only in the case of the quantum states associated with the algebra B(H), but in the more general case of states of a finite-dimensional $C^{*}$-algebra A. This framework allows us to deal with classical and quantum states with the same formalism because the states of a finite-dimensional, Abelian $C^{*}$-algebra A are in one-to-one correspondence with the probability distributions on a finite-outcome space. Furthermore, it nicely fits into the recently developed groupoidal approach to quantum theories developed in [32,33,34,35,36,37].

Then, in Section 6, we will take inspiration from Uhlmann’s geometric construction of the Bures–Helstrom metric tensor (see [38,39,40,41,42,43,44,45,46,47]) to show that the Riemannian geometries on the states associated with the Jordan product may be realized as “projected shadows” of the Riemannian geometries of suitable spheres in suitable Hilbert spaces by means of the GNS construction.

The geometrical picture that will eventually emerge from the present work is that the Jordan product derived from the associative product in A generates Riemann metric tensors on the manifolds of states on A that are associated with the action of the group G of invertible elements in A. In particular, in Section 5, Section 7, and Section 8 respectively, these Riemannian metric tensors are shown to coincide with the Fisher–Rao metric tensor when A is Abelian, with the Fubini–Study metric tensor when A=B(H) and we consider the manifold of pure quantum states (rank-one projectors in B(H)), and with the Bures–Helstrom metric tensor when A=B(H) and we consider the manifold of invertible quantum states. Consequently, with the help of some imagination, we may interpret all these seemingly different metric tensors as being different faces of the same object, namely, the contravariant tensor R determined by the Jordan product. Section 9 contains some concluding remarks.

## 2. Geometrical Aspects of Positive Linear Functionals and States

Let A be a finite-dimensional, unital $C^{*}$-algebra in which the involution is denoted by † and unit by I. We refer to [48,49,50] for the basic definitions concerning $C^{*}$-algebras. Let $A_{sa}$ be the self-adjoint part of A. The associative product of A gives rise to a commutative product {,} and to a non-commutative product [[,]] on $A_{sa}$ by setting
(1)$$\begin{array}{cc} \left\{a,b\right\} & \;:=\;\frac{1}{2}\;\left(\mathrm{ab}+\mathrm{ba}\right)\\ \left[[a,b]\right] & \;:=\;\frac{1}{2ı}\;\left(\mathrm{ab}-\mathrm{ba}\right), \end{array}$$
where ı is the imaginary unit. Note that both {,} and [[,]] are non-associative, and [[,]] satisfies the Jacobi identity. These two products make $A_{sa}$ into a Banach–Lie–Jordan algebra (see [51,52,53,54,55,56]). In particular, [[a,·]] defines a derivation of {,} for every $a\in A_{sa}$. Furthermore, since the Lie product [[,]] makes $A_{sa}$ into a Banach–Lie algebra, it is possible to show that there is a Banach–Lie group U of which $\left(A_{sa},\left[\left[,\right]\right]\right)$ is the Banach–Lie algebra. The group U is just the group of unitary elements in A, and is a subgroup of the Banach–Lie group G of invertible elements in A (see [57]).

Let V be the self-adjoint part of the Banach dual $A^{*}$ of A, that is, the set of all the linear functionals ξ on A such that
(2)$$\xi \left(a^{†}\right)\;=\;\bar{\xi (a)}\;\forall \;a\in \;A,$$
where $\bar{\cdot }$ denotes complex conjugation. A linear functional $\omega \in V\subset A^{*}$ is called positive if
(3)$$\omega ({\mathrm{aa}}^{†})\;\geq \;0\;\;\forall \;\;a\;\in \;A.$$
A positive linear functional ω is called faithful if
(4)$$\omega ({\mathrm{aa}}^{†})\;=\;0\;\;\;⟺\;\;\;a\;=\;0.$$
The space of positive linear functionals on A (excluding the null functional) is denoted by P, while $P_{+}$ denotes the space of faithful, positive linear functionals, which is an open submanifold of V.

For future reference, we need to briefly recall the so-called Gelfand–Naimark–Segal (GNS) construction associated with a positive linear functional ω (see [48,50,58] for more details). Given ω, we define the set
(5)$$N_{\omega }\;:=\;\left\{a\in A\;|\;\;\omega (a^{†}\;a)\;=\;0\right\}.$$
This is a left ideal in A called the ***Gel’fand ideal*** of ω. Then, we consider the bilinear form on A given by
(6)$${\left(a,b\right)}_{\omega }\;:=\;\omega \left(a^{†}\;b\right).$$
It is easily seen that ${\left(,\right)}_{\omega }$ is a pre-inner product that descends to the quotient
(7)$$H_{\omega }\;=\;A/N_{\omega }.$$
By completing $H_{\omega }$ with respect to ${\left(,\right)}_{\omega }$, we obtain a complex Hilbert space $H_{\omega }$ the elements of which are written as $\psi_{a}$ to emphasize that they are associated with (equivalence classes of) elements of A. The Hilbert space product on $H_{\omega }$ is written as 〈,〉, and there is a representation $r_{\omega }$ of A in $B(H_{\omega })$ given by
(8)$$\left(r_{\omega }\left(a\right)\right)\left(\psi_{b}\right)\;:=\;\psi_{ab}.$$
The vector $\psi_{I}$ is cyclic with respect to $r_{\omega }$ and separating for the commutant of $r_{\omega }\left(A\right)$ in $B(H_{\omega })$ (see prop. 2.5.3 of [49]). Moreover, every vector in $H_{\omega }$ gives rise to a positive linear functional on A by means of
(9)$$\omega_{a}\left(b\right)\;:=\;\langle \psi_{a}|r_{\omega }\left(b\right)|\psi_{a}\rangle .$$
If ω is faithful, then $N_{\omega }=\left\{0\right\}$ and $H_{\omega }=A$. Therefore, in the finite-dimensional case, $H_{\omega }$ coincides with A and $r_{\omega }$ becomes the left regular representation $a\mapsto L_{a}$ of A on itself.

**Proposition** **1.**
*Let ω be a positive linear functional on A, and let a∈A be an element of the Gel’fand ideal $N_{\omega }$ (see Equation (Equation 5)). Then, we have*
(10)$$\omega \left(a^{†}\;b\right)\;=\;\omega \left(b\;a\right)\;=\;0$$
*for all $b\in A_{sa}$.*


**Proof.** We just need to apply the Cauchy–Schwarz inequality (see prop. 2.3.10.b of [49]) to the positive-semidefinite sesquilinear form defined by ω in Equation (Equation 6). Specifically, we have
(11)$${\left|{\left(a,b\right)}_{\omega }\right|}^{2}\;=\;{\left|\omega (a^{†}\;b)\right|}^{2}\;\leq \;\omega \left(a^{†}a\right)\;\omega \left(b^{2}\right)\;=\;0,$$
where, in the last equality, we used the fact that a is in $N_{\omega }$.

The space P is not a smooth manifold in the usual sense of differential geometry (see [59,60] for the appropriate definition of smooth manifold). However, there is a linear left action α of the Lie group G of invertible elements in A on the space V given by (see [29,30,31])
(12)$$\alpha \left(g,\xi \right)\;\equiv \;\xi_{g}\;\;:\;\;\;\xi_{g}\left(a\right)\;:=\;\xi \left(g^{†}\;a\;g\right)\;\;\forall \;\;a\;\in \;A_{sa}.$$
This action preserves P, and every orbit of α is a smooth submanifold of V. That is, even though P itself does not have a smooth structure, it is stratified by the orbits of this action all of which are homogeneous spaces and hence smooth manifolds; in fact, the manifold structure of these homogeneous spaces coincides with that induced by the inclusion into V as the action of G is defined on all of V. The top stratum, i.e., that of maximal dimension, is easily seen to be the space $P_{+}$ of faithful, positive linear functionals.

Recall that every $a\in A_{sa}$ may be identified with a real-valued, linear function on V, that we denote by $l_{a}$, by means of the expression
(13)$$l_{a}\left(\xi \right)\;:=\;\xi \left(a\right).$$
Since V is a finite-dimensional Banach space, the map $a\mapsto l_{a}$ is an isomorphism between $A_{sa}$ and $V^{*}=A_{sa}^{**}$, and thus the differential of the linear functions on V associated with elements in $A_{sa}$ generate the cotangent space $T_{\xi }^{*}V$ at each ξ.

Now, given $a\in A_{sa}$, we introduce the vector fields $X_{a}$ and $Y_{a}$ given by
(14)$$\begin{array}{cc} X_{a}\left(l_{b}\right) & \;=\;l_{\left[[a,b]\right]}\\ Y_{a}\left(l_{b}\right) & \;=\;l_{\left\{a,b\right\}}\;. \end{array}$$
For reasons that will be clear later, we call $Y_{a}$ a gradient vector field, and $X_{a}$ a Hamiltonian vector field. Next, we define the vector field
(15)$$V_{ab}\;:=\;Y_{a}+X_{b}\;.$$
A direct computation based on Equation (Equation 14) shows that the Lie bracket [,] between $V_{ab}$ and $V_{cd}$ reads
(16)$$\left[V_{ab},\;V_{cd}\right]\;=\;Y_{\left[[a,d]]+[[b,c]\right]}+X_{\left[[c,a]]+[[b,d]\right]}\;.$$
In particular, it follows that the Hamiltonian vector fields ${\left\{X_{a}\right\}}_{a\in A_{sa}}$ provide an anti-representation of the Lie algebra u of the unitary group U of A. According to what will be proved below, the left action of U generated by the Hamiltonian vector fields is just α restricted to U.

We will now prove that the vector fields $V_{ab}$ are the fundamental vector fields of the action α. Specifically, we consider an element g∈G, and write it as
(17)$$g=e^{\frac{1}{2}\left(a+ıb\right)}$$
with $a,b\in A_{sa}$. This is always possible because $A=A_{sa}⊕ıA_{sa}$, where ı is the imaginary unit, is the Lie algebra of the Lie group G of invertible elements in A. Then, we may consider the smooth curve
(18)$$g\left(t\right)=e^{\frac{t}{2}\left(a+ıb\right)}$$
starting at g(0)=I, and compute the fundamental vector field *F* of the action α associated with g(t). This vector field is defined as the infinitesimal generator of the one-parameter group of diffeomorphisms of V generated by g(t) by means of α (see [59] (p. 331)). Specifically, it is
(19)$$\begin{array}{cc} \langle dl_{c}\left(\omega \right),F\left(\xi \right)\rangle & \;=\;\frac{d}{dt}\;{\left(l_{c}\left(\alpha (g(t),\;\xi )\right)\right)}_{t=0}\;=\;\\ & \;=\;\frac{d}{dt}\;{\left(\xi \left(e^{\frac{t}{2}\left(a-ıb\right)}\;c\;e^{\frac{t}{2}\left(a+ıb\right)}\right)\right)}_{t=0}\;=\;\\ & \;=\;\xi \left(\{a,c\}+[[b,\;c]]\right), \end{array}$$
for all ξ∈V. Comparing Equation (Equation 19) with Equations (Equation 15) and (Equation 14), we conclude that $F=V_{ab}$ as claimed.

Let us fix an orbit O⊂P of G. The tangent space $T_{\omega }O$ is thus identified with the subspace of $V≅T_{\omega }V$ written as
(20)$$T_{\omega }O\;≅\;\left\{\omega_{ab}\in V\;|\;\;\;\omega_{ab}\left(c\right)\;=\;\omega \left(\left\{a,c\right\}+\left[\left[b,c\right]\right]\right)\;\forall \;c\in A_{sa}\right\}.$$
In this work, the symbol ≅ denotes the identification of two different sets. Then, the cotangent space $T_{\omega }^{*}O$ is isomorphic with $T_{\omega }^{*}V/Ann\left(T_{\omega }O\right)$ where $Ann(T_{\omega }O)$ is the annihilator of $T_{\omega }O$ inside $T_{\omega }^{*}V$. From the practical point of view, we define the functions
(21)$$l_{b}^{+}\;:=\;i^{*}\;l_{b}$$
for every $b\in A_{sa}$, where *i* is the canonical immersion of the orbit O⊂P in V, and we obtain that the cotangent vector $dl_{b}^{+}\left(\omega \right)$ at every ω∈O is identified with b. Clearly, since the set ${\left\{dl_{b}\left(\xi \right)\right\}}_{b\in A_{sa}}$ is an overcomplete basis for $T_{\xi }^{*}V$ for every ξ∈V, we have that the set ${\left\{dl_{b}^{+}\left(\omega \right)\right\}}_{b\in A_{sa}}$ is an overcomplete basis for $T_{\omega }^{*}O$ for every ω∈O.

Now, we will pass from positive linear functional, to states. A positive linear functional ρ is called a state if
(22)ρ(I)=1,
where I is the identity element in A. The space of states S is the convex body in P which is the intersection of P with the affine hyperplane determined as the inverse image of 1 through the linear function $l_{I}$, with I∈A being the identity element. Consequently, if O is an orbit of G in P through ω, we may consider the inverse image of 1 through the function $l_{I}^{+}$ and obtain a smooth manifold, denoted by $O_{1}$, of states as a closed submanifold of O. Clearly, we may do that for every orbit O in P, and thus we obtain a stratification of S into the disjoint union of smooth manifolds, where the top stratum, denoted by $S_{+}$, is the space of faithful states. Note that some of these manifolds can be degenerate, i.e., points. In particular, this happens when A is Abelian and $O_{1}$ contains a pure state (recall that pure states are the extremal points of S). In the following, whenever we consider a manifold $O_{1}$ of states, we will always implicitly assume that $O_{1}$ is not a single point.

Concerning pure states, it is worth mentioning that, according to [61], the functional representation of a commutative $C^{*}$-algebra in terms of complex-valued functions on the space of pure states may be extended to any noncommutative $C^{*}$-algebra by looking at the space of pure states as a bundle of Kähler manifolds, and using the Kähler metric to define a noncommutative product between complex-valued functions on the pure states.

We will now see how the group G acts on every $O_{1}$ making it a homogeneous space. To this scope, we first note that, if ρ is a state sitting inside O⊂P, then α(g,ρ) is in general not a state. From the infinitesimal point of view, this is related to the fact that the gradient vector fields on O do not preserve $O_{1}$ because $Y_{a}\left(l_{I}^{+}\right)$ in general does not vanish. However, if we set
(23)$$\tilde{Y_{a}}\;:=\;Y_{a}-l_{a}^{+}Y_{I},$$
then $\tilde{Y_{a}}\left(l_{I}^{+}\right)\;=\;Y_{a}\left(l_{I}^{+}\right)-l_{a}^{+}Y_{I}\left(l_{I}^{+}\right)\;=\;0$, and thus $\tilde{Y_{a}}$ is tangent to $O_{1}$. This means that there is a vector field $Y_{a}$ on $O_{1}$ which is $i_{1+}$-related to $\tilde{Y_{a}}$, where $i_{1+}:O_{1}⟶O$ is the canonical immersion map given by identification. Furthermore, every Hamiltonian vector field $X_{a}$ is tangent to $O_{1}$, and we denote by $X_{a}$ the vector field on $O_{1}$ which is $i_{1+}$-related with $X_{a}$. For reasons that will be clear later, we call $Y_{a}$ a gradient vector field, and $X_{a}$ a Hamiltonian vector field.

Now, we define the vector fields ${\left\{Y_{ab}\right\}}_{a,b\in A_{sa}}$ by means of
(24)$$Y_{ab}\;:=\;Y_{a}+X_{b}.$$
Quite interestingly, a direct computation shows that
(25)$$\left[\tilde{Y_{a}}+X_{b},\;\tilde{Y_{c}}+X_{d}\right]\;=\;{\tilde{Y}}_{\left[[a,d]]+[[b,c]\right]}+X_{\left[[c,a]]+[[b,d]\right]}\;,$$
which means that we also have
(26)$$\left[Y_{ab},\;Y_{cd}\right]\;=\;Y_{\left[[a,d]]+[[b,c]\right]}+X_{\left[[c,a]]+[[b,d]\right]}\;.$$
Comparing Equation (Equation 26) with Equation (Equation 16), we conclude that the vector fields ${\left\{Y_{ab}\right\}}_{a,b\in A_{sa}}$ provide a representation of the Lie algebra g of G which is tangent to $O_{1}$. Furthermore, if we define the map $\Phi :O_{1}⟶O_{1}$ given by (see also [29,30,31])
(27)$$\Phi \left(g,\rho \right)\;\equiv \;\rho_{g}\;\;:\;\;\;\rho_{g}\left(a\right)\;:=\;\frac{\left(\alpha (g,\rho ))(a\right)}{\left(\alpha (g,\rho ))(I\right)}\;=\;\frac{\rho (g^{†}\;a\;g)}{\rho (g^{†}\;g)}\;\;\forall \;\;a\;\;\in A_{sa},$$
it is not hard to show that it is a left action of G on $O_{1}$ which is transitive (essentially because α is transitive on O). In particular, the space $S_{+}$ of faithful states is an orbit of G. The flow of the vector field $Y_{ab}$ is just Φ(g(t),ρ), where g(t) is defined as in Equation (Equation 18), and thus the fundamental vector fields of Φ are precisely the $Y_{ab}$’s.

Note that the map Φ is well-defined because the denominator term is always strictly positive since g is an invertible element. However, note that Φ does not preserve the convex structure of S, that is, we have
(28)$$\Phi \left(g,\;\lambda \;\rho_{1}+\left(1-\lambda \right)\rho_{2}\right)\;\neq \;\lambda \;\Phi \left(g,\;\rho_{1}\right)+\left(1-\lambda \right)\;\Phi \left(g,\;\rho_{2}\right)$$
in general.

Let us now fix the orbit $O_{1}\subset S$. Defining the function
(29)$$e_{a}\;:=\;i_{1+}^{*}l_{a}^{+}\;=\;i_{1+}^{*}\;i^{*}l_{a}$$
for every $a\in A_{sa}$, it is immediate to check that
(30)$$\begin{array}{cc} X_{a}\left(e_{b}\right) & \;=\;e_{\left[[a,b]\right]}\\ Y_{a}\left(e_{b}\right) & \;=\;e_{\left\{a,b\right\}}-\;e_{a}\;e_{b}. \end{array}$$
The tangent space at $\rho \in O_{1}$ is identified with the subspace of $V≅T_{\rho }V$ written as
(31)$$T_{\rho }O_{1}\;≅\;\left\{\rho_{ab}\in V\;|\;\;\;\rho_{ab}\left(c\right)\;=\;\rho \left(\left\{a,c\right\}+\left[\left[b,c\right]\right]\right)-\rho \left(a\right)\;\rho \left(c\right)\;\forall \;c\in A_{sa}\right\},$$
while the cotangent space $T_{\rho }^{*}O_{1}$ is isomorphic with $T_{\rho }^{*}V/Ann\left(T_{\rho }O_{1}\right)$ where $Ann(T_{\rho }O_{1})$ is the annihilator of $T_{\rho }O_{1}$ inside $T_{\rho }^{*}V$. From the practical point of view, given $\rho \in O_{1}$, just as it happens for the orbit O⊂P, we obtain that the cotangent vector $de_{b}\left(\rho \right)$ is identified with b. Clearly, since the set ${\left\{dl_{a}\left(\xi \right)\right\}}_{a\in A_{sa}}$ is an overcomplete basis for $T_{\xi }^{*}V$ for every ξ∈V, we have that the set ${\left\{de_{b}\left(\rho \right)\right\}}_{b\in A_{sa}}$ is an overcomplete basis for $T_{\rho }^{*}O_{1}$ for every $\rho \in O_{1}$.

**Remark** **1.**
*An identification similar to that given in Equation (Equation 31) (with b=0) may be found also in [5], under the name of e-representation, and in [62] for faithful states. However, in these works, the authors consider only the case A=B(H) (with dim(H)<∞) so that they identify the space of faithful states $S_{+}$ with the space of faithful density operators in B(H) by means of the isomorphism between B(H) and its dual induced by the trace on H, and no mention is made of the gradient and Hamiltonian vector fields nor of the associated action of G on $S_{+}$. On the other hand, here we want to stress that the identification of $T_{\rho }O_{1}$ with a linear subspace of V given by Equation (Equation 31) works for every orbit $O_{1}\subset S$ and it is part of the “internal geometry” of the space of states of A.*


## 3. From the Jordan Product to Riemannian Geometries

We will now exploit the Jordan–Lie-algebra structure of $A_{sa}$ introduced above to obtain geometric tensor fields on V, specifically, we obtain a symmetric, contravariant bivector field R associated with the Jordan product {,}, and a Poisson bivector field Λ associated with the Lie product [[,]] on $A_{sa}$. This is the generalization to a generic (finite-dimensional) $C^{*}$-algebra A of what is done in [19,20,21,22,23,24,25] for the specific case A=B(H) for a finite-dimensional Hilbert space H. Then, we will show how the manifolds of positive linear functionals introduced in the previous section may be interpreted as a sort of analogs of symplectic leaves for the symmetric tensor R in a sense that will be specified later. This will allow us to define Riemannian geometries on the orbits of G in P that will be studied in some detail.

In order to define R and Λ, we recall that the differentials of the linear functions $l_{a}$ with $a\in A_{sa}$ generate the cotangent space $T_{\xi }^{*}V$ at every ξ∈V, so that we may set
(32)$$\left(R(dl_{a},dl_{b})\right)\left(\xi \right)\;:=\;l_{\left\{a,b\right\}}\left(\xi \right)\;=\;\xi \left(\left\{a,b\right\}\right),$$
(33)$$\left(\Lambda (dl_{a},dl_{b})\right)\left(\xi \right)\;:=\;l_{\left[[a,b]\right]}\left(\xi \right)\;=\;\xi \left(\left[\left[a,b\right]\right]\right),$$
and extend these objects by linearity obtaining two contravariant tensor fields
(34)$$\left(R(df_{1},df_{2})\right)\left(\xi \right)\;:=\;\xi \left(\{df_{1}\left(\xi \right),\;df_{2}\left(\xi \right)\}\right),$$
(35)$$\left(\Lambda (df_{1},df_{2})\right)\left(\xi \right)\;:=\;\xi \left([\left[df_{1}\left(\xi \right),\;df_{2}\left(\xi \right)\right]]\right).$$
The antisymmetry of [[,]] implies that Λ is antisymmetric, while the symmetricity of {,} implies that R is symmetric. Moreover, note that both tensors have non-constant rank, and Λ=0 if A is Abelian.

If A=B(H) for some finite-dimensional Hilbert space H, it is a matter of direct inspection to show that tensor fields R and Λ defined above coincide with those introduced in [19,20,21,22,23,24,25].

The Lie algebra of the unitary group U may be identified with the space $A_{sa}$ of self-adjoint elements in A (see Equation (Equation 17)), and thus Λ may be interpreted as the Kostant–Kirillov–Souriau Poisson tensor associated with the coadjoint action of the unitary group U. Since Λ is a Poisson tensor, we may introduce the Hamiltonian vector field $X_{f}$ associated with a smooth function *f* on V by means of Λ by setting
(36)$$X_{f}\;:=\;\Lambda \left(df,\;\cdot \right)\;.$$
In particular, it is immediate to check that the Hamiltonian vector field associated with the linear function $l_{a}$ (with $a\in A_{sa}$) coincides with the fundamental vector field $X_{a}=V_{0a}$ for α introduced in the previous section. Analogously, we may (improperly) define the gradient vector field $Y_{f}$ associated with a smooth function *f* on V by means of R by setting
(37)$$Y_{f}\;:=\;R\left(df,\;\cdot \right).$$
Again, it is immediate to check that the gradient vector field associated with the linear function $l_{a}$ (with $a\in A_{sa}$) coincides with the fundamental vector field $Y_{a}=V_{a0}$ for α introduced in the previous section. This gives an intimate connection between the tensor field Λ and R and the action α of G on V. In particular, the Hamiltonian vector fields $X_{a}$ generate the action of the unitary group U⊂G and the orbits of this action, which are embedded, compact submanifolds of V because U is a compact group (only in finite dimensions), are the leaves of the symplectic foliation associated with the Poisson tensor Λ. When A coincides with the $C^{*}$-algebra B(H) of linear operators on the finite-dimensional, complex Hilbert space H, it is not hard to see that the orbit through ξ∈V is in one-to-one correspondence with the set of self-adjoint operators on H that are isospectral with the self-adjoint operator $\tilde{\xi }$, which is uniquely associated to ξ by means of the isomorphism between $A_{sa}$ and V induced by the trace on H. When A is Abelian, then Λ=0 and the action of U on V is trivial.

The orbits of U are such that the Poisson tensor Λ is invertible on them and thus gives rise to a symplectic form on every orbit. We may try to obtain a similar construction for the tensor field R, that is, we may try to find suitable submanifolds of V on which R is invertible. In a certain sense, we are looking for analogs, for R, of what would be the symplectic leaves of Λ.

Quite interestingly, we will see that every orbit O of positive linear functionals provides an example of such an analog of a symplectic leaf. In particular, we will see that the inverse G of R on O determines a Riemannian metric tensor, and compute its associated covariant derivative, sectional curvature, and Riemann tensor. Then, we will study the Riemannian geometry on the orbit $O_{1}\subset O$ of states arising from the canonical immersion $i_{1+}:O_{1}⟶O$ given by the identification map.

Let us fix the orbit O⊂P. It is an immersed submanifold of V, consequently, the set ${\left\{dl_{a}^{+}\left(\omega \right)\right\}}_{a\in A_{sa}}$, where $l_{a}^{+}=i^{*}l_{a}$, is an overcomplete basis for the cotangent space $T_{\omega }O$. Therefore, we may define the symmetric, (0,2) tensor field *R* on O by setting
(38)$$\left(R(dl_{a}^{+},dl_{b}^{+})\right)\left(\omega \right)\;:=\;\omega \left(\left\{a,b\right\}\right),$$
and then extend by linearity just as we did for the definition of R. By construction, we have that
(39)$$R\left(i^{*}\theta_{1},i^{*}\theta_{2}\right)\;=\;i^{*}\;\left(R(\theta_{1},\theta_{2})\right)$$
for all 1-forms $\theta_{1},\theta_{2}$ on V. The tensor *R* may be thought of as the restriction of R to O.

**Proposition** **2.**
*The contravariant tensor R is symmetric, invertible and positive.*


**Proof.** By definition of *R*, a cotangent vector $dl_{c}^{+}\left(\omega \right)$ at ω∈O is such that
(40)$$R_{\omega }\left(dl_{c}^{+}\left(\omega \right),dl_{c}^{+}\left(\omega \right)\right)\;=\;\omega \left(c^{2}\right)\;\geq \;0$$
because ω is a positive linear functional. Recalling the definition of the GNS ideal $N_{\omega }$ of ω given in Equation (Equation 5), and recalling that c is self-adjoint, it is immediate to see that the equality in the previous equation holds if and only if $dl_{c}^{+}\left(\omega \right)=c\;\in \;N_{\omega }\;\cap \;A_{sa}\;\forall \;\omega \in O$. Moreover, given any tangent vector $v_{\omega }$ at ω∈O, we may find a fundamental vector field $V_{ab}$ such that $v_{\omega }=V_{ab}\left(\omega \right)$ because O is a homogeneous space for G. Therefore, we have
(41)$$\langle dl_{c}^{+}\left(\omega \right),v_{\omega }\rangle \;=\;\langle dl_{c}^{+}\left(\omega \right),V_{ab}\left(\omega \right)\rangle \;=\;\omega \left(\left\{a,c\right\}\right)+\omega \left(\left[\left[b,c\right]\right]\right)\;=\;0,$$
where the last equality follows from the fact that c is in $N_{\omega }$ and from Proposition 1. Then, since the tangent vector $v_{\omega }$ was arbitrary, we conclude that the cotangent vector $dl_{c}^{+}\left(\omega \right)$ must be the zero cotangent vector, and thus it follows that *R* is positive and invertible on O⊂P.

Because of Proposition 2, the covariant tensor
(42)$$G\;:=\;R^{-1}$$
is a Riemannian metric tensor on the orbit O⊂P. We can immediately compute the gradient vector field $W_{a}$ associated with the function $l_{a}^{+}$ by means of G. In order to characterize $W_{a}$, it is sufficient to obtain its action on all the functions $l_{b}^{+}$ with $b\in A_{sa}$ because the set ${\left\{dl_{b}^{+}\left(\omega \right)\right\}}_{b\in A_{sa}}$ is an overcomplete basis for $T_{\omega }^{*}O$ for every ω∈O. By definition of gradient vector field, we have
(43)$$W_{a}\left(l_{b}^{+}\right)\;=\;R\left(dl_{a}^{+},dl_{b}^{+}\right)\;=\;i_{+}^{*}\;\left(R(dl_{a},dl_{b})\right)\;=\;i_{+}^{*}\left(Y_{a}\left(l_{b}\right)\right),$$
from which we obtain that
(44)$$W_{a}\;=\;Y_{a}$$
with $Y_{a}$ the fundamental vector field $V_{a0}$ of the action α of G. Consequently, we have
(45)$$G\left(Y_{a},Y_{b}\right)\;=\;l_{\left\{a,b\right\}}^{+}\;,$$
and the fact that $Y_{a}$ is the gradient vector field associated with $l_{a}^{+}$ by means of G explains why we already called it a gradient vector field when we defined it in Section 2. Furthermore, since the set ${\left\{dl_{b}^{+}\left(\omega \right)\right\}}_{b\in A_{sa}}$ is an overcomplete basis for $T_{\omega }^{*}O$ for every ω∈O, the set ${\left\{Y_{b}\left(\omega \right)\right\}}_{b\in A_{sa}}$ is an overcomplete basis for $T_{\omega }O$ for every ω∈O.

By directly applying the definition of a gradient vector field, we have
(46)$$G\left(Y_{a},X_{b}\right)\;=\;dl_{a}^{+}\left(X_{b}\right)\;=\;X_{b}l_{a}^{+}\;=\;l_{\left[[b,a]\right]}^{+}\;.$$
Furthermore, we have
(47)$$\begin{array}{cc} X_{a}\left(G(Y_{b},Y_{c})\right) & \;=\;\left(L_{X_{a}}G\right)\left(Y_{b},Y_{c}\right)+G\left(\left[X_{a},Y_{b}\right],Y_{c}\right)+G\left(Y_{b},\left[X_{a},Y_{c}\right]\right)\;=\;\\ & \;=\;\left(L_{X_{a}}G\right)\left(Y_{b},Y_{c}\right)+G\left(Y_{\left[[a,b]\right]},Y_{c}\right)+G\left(Y_{b},Y_{\left[[a,c]\right]}\right)\;=\;\\ & \;=\;\left(L_{X_{a}}G\right)\left(Y_{b},Y_{c}\right)+l_{\left\{[[a,b]],c\right\}}^{+}+l_{\left\{b,[[a,c]]\right\}}^{+} \end{array}$$
where $L_{X_{a}}$ denotes the Lie derivative (see [59,60]), and where we used Equations (Equation 45) and (Equation 16). However, it also holds
(48)$$X_{a}\left(G(Y_{b},Y_{c})\right)\;=\;X_{a}\;l_{\left\{b,c\right\}}^{+}\;=\;l_{\left\{[[a,b]],c\right\}}^{+}+l_{\left\{b,[[a,c]]\right\}}^{+},$$
where we used Equation (Equation 45), the first equality in Equation (Equation 14), and the fact that the Lie product is a derivation of the Jordan product. Therefore, comparing the previous two equations, and recalling that the Jordan product is symmetric, we obtain
(49)$$L_{X_{a}}G\;=\;0$$
for all $a\in A_{sa}$. This means that the metric G is invariant under the action of the unitary group U defined by the Hamiltonian vector fields associated with the functions $l_{a}^{+}$ with $a\in A_{sa}$.

Regarding the evaluation of G on Hamiltonian vector fields, we can say the following. First of all, we note that the set ${\left\{Y_{b}\left(\omega \right)\right\}}_{b\in A_{sa}}$ is an overcomplete basis for $T_{\omega }O$ for every ω∈O, therefore, given the Hamiltonian vector field $X_{b}$, we can express the tangent vector $X_{b}\left(\omega \right)$ in terms of gradient tangent vectors, that is, we have
(50)$$X_{b}\left(\omega \right)\;=\;Y_{B_{\omega }^{b}}\left(\omega \right),$$
where $B_{\omega }^{b}$ is an element of $A_{sa}$ that depends on both b and ω. Specifically, $B_{\omega }^{b}$ is the element in $A_{sa}$ such that
(51)$$\omega \left(\left[\left[b,c\right]\right]\right)\;=\;\omega \left(\left\{B_{\omega }^{b},c\right\}\right)$$
for all $c\in A_{sa}$. In general, $B_{\omega }^{b}$ is not unique, but it is defined up to an element in the isotropy algebra $g_{\omega }$ of ω. However, since the gradient tangent vector $Y_{a}\left(\omega \right)$ vanishes for every $a\in A_{sa}\cap g_{\omega }$, we conclude that the non-uniqueness of $B_{\omega }^{b}$ does not affect $Y_{B_{\omega }^{b}}\left(\omega \right)$. Consequently, we may write
(52)$$\begin{array}{cc} \left(G(X_{a},X_{b})\right)\left(\omega \right) & \;=\;G_{\omega }\left(X_{a}\left(\omega \right),X_{b}\left(\omega \right)\right)\;=\;\\ & \;=\;G_{\omega }\left(X_{a}\left(\omega \right),Y_{B_{\omega }^{b}}\left(\omega \right)\right)\;=\;\\ & \;=\;l_{\left[\left[a,B_{\omega }^{b}\right]\right]}^{+}\left(\omega \right)\;=\;\omega \left([\left[a,B_{\omega }^{b}\right]]\right)\;. \end{array}$$
Note that, unlike Equations (Equation 45) and (Equation 46), Equation (Equation 52) expresses a pointwise relation.

Now, we will compute the covariant derivative associated with the metric G. By applying the Koszul formula (see [60] (thm. 4.3.1)), we obtain
(53)$$\begin{array}{cc} \nabla_{Y_{a}}Y_{b}\left(l_{c}^{+}\right) & \;=\;G(\nabla_{Y_{a}}Y_{b},Y_{c})\;=\;\\ & \;=\;\frac{1}{2}\left(Y_{a}\left(G(Y_{b},Y_{c})\right)+Y_{b}\left(G(Y_{a},Y_{c})\right)-Y_{c}\left(G(Y_{a},Y_{b})\right)+\right.\\ & \;\;\;\left.+\;G\left(\left[Y_{a},Y_{b}\right],Y_{c}\right)-G\left(\left[Y_{a},Y_{c}\right],Y_{b}\right)-G\left(\left[Y_{b},Y_{c}\right],Y_{a}\right)\right)\;=\;\\ & \;=\;\frac{1}{2}\left(l_{\left\{a,\{b,c\}\right\}}^{+}+l_{\left\{b,\{a,c\}\right\}}^{+}-l_{\left\{c,\{a,b\}\right\}}^{+}+l_{\left[[[[b,a]],c]\right]}^{+}-l_{\left[[[[c,a]],b]\right]}^{+}-l_{\left[[[[c,b]],a]\right]}^{+}\right), \end{array}$$
where we used Equations (Equation 46) and (Equation 48). Expanding the Lie and Jordan products according to Equation (Equation 1), some tedious but simple manipulations show that
(54){a,{b,c}}+{b,{a,c}}−[[[[c,a]],b]]−[[[[c,b]],a]]=2{{a,b},c},
from which it follows that
(55)$$\begin{array}{cc} \nabla_{Y_{a}}Y_{b}\left(l_{c}^{+}\right) & \;=\;\frac{1}{2}\left(l_{\left\{c,\{a,b\}\right\}}^{+}+l_{\left[[[[b,a]],c]\right]}^{+}\right)\;=\;\\ & \;=\;\frac{1}{2}\left(Y_{\left\{a,b\right\}}\left(l_{c}^{+}\right)-X_{\left[[a,b]\right]}\left(l_{c}^{+}\right)\right), \end{array}$$
where we used Equation (Equation 14). Eventually, we may write the covariant derivative of a gradient vector field with respect to another gradient vector field as
(56)$$\nabla_{Y_{a}}Y_{b}\;=\;\frac{1}{2}\left(Y_{\left\{a,b\right\}}-X_{\left[[a,b]\right]}\right)$$

At this point, we may compute the Riemann curvature tensor of G as
(57)$$\begin{array}{cc} G\left(R_{G}\left(Y_{a},Y_{b}\right)Y_{c},Y_{d}\right) & \;=\;G\left(\nabla_{Y_{a}}\nabla_{Y_{b}}Y_{c},Y_{d}\right)-G\left(\nabla_{Y_{b}}\nabla_{Y_{a}}Y_{c},Y_{d}\right)-G\left(\nabla_{\left[Y_{a},Y_{b}\right]}Y_{c},Y_{d}\right)\;. \end{array}$$
A direct application of Equation (Equation 56) gives
(58)$$\begin{array}{cc} \nabla_{Y_{a}}\nabla_{Y_{b}}Y_{c} & \;=\;\frac{1}{4}\left(Y_{\left\{a,\{b,c\}\right\}}-X_{\left[[a,\{b,c\}]\right]}-2\nabla_{Y_{a}}X_{\left[[b,c]\right]}\right)\\ \nabla_{Y_{b}}\nabla_{Y_{a}}Y_{c} & \;=\;\frac{1}{4}\left(Y_{\left\{b,\{a,c\}\right\}}-X_{\left[[b,\{a,c\}]\right]}-2\nabla_{Y_{b}}X_{\left[[a,c]\right]}\right)\\ \nabla_{\left[Y_{a},Y_{b}\right]}Y_{c} & \;=\;\nabla_{X_{\left[[b,a]\right]}}Y_{c}\;=\;\nabla_{Y_{c}}X_{\left[[b,a]\right]}+Y_{\left[[[[b,a]],c]\right]}\;, \end{array}$$
and thus
(59)$$\begin{array}{cc} G\left(R_{G}\left(Y_{a},Y_{b}\right)Y_{c},Y_{d}\right) & \;=\;\frac{1}{4}G\left(X_{\left[[b,\{a,c\}]\right]}-X_{\left[[a,\{b,c\}]\right]}-Y_{\left\{b,\{a,c\}\right\}}+X_{\left[[b,\{a,c\}]\right]}+2Y_{\left[[[[a,b]],c]\right]},Y_{d}\right)+\\ & \;\;\;\;+\frac{1}{2}G\left(\nabla_{Y_{b}}X_{\left[[a,c]\right]}-\nabla_{Y_{a}}X_{\left[[b,c]\right]}+2\nabla_{Y_{c}}X_{\left[[a,b]\right]},Y_{d}\right). \end{array}$$
In order to compute the scalar products in the last line of the previous equation, we start noting that, since ∇ is compatible with G, we have
(60)$$Y_{a}\left(G(X_{b},Y_{c})\right)\;=\;G\left(\nabla_{Y_{a}}X_{b},Y_{c}\right)+G\left(X_{b},\nabla_{Y_{a}}Y_{c}\right)$$
for every $Y_{a},X_{b},Y_{c}$, which implies
(61)$$\begin{array}{cc} G\left(\nabla_{Y_{a}}X_{b},Y_{c}\right) & \;=\;l_{\left\{a,[[b,c]]\right\}}^{+}-\frac{1}{2}G\left(X_{b},Y_{\left\{a,c\right\}}\right)+\frac{1}{2}G\left(X_{b},X_{\left[[a,c]\right]}\right)\;=\;\\ & \;=\;\frac{1}{2}l_{\left\{a,[[b,c]]\right\}}^{+}-\frac{1}{2}l_{\left\{c,[[b,a]]\right\}}^{+}+\frac{1}{2}G\left(X_{b},X_{\left[[a,c]\right]}\right)\;=\;\\ & \;=\;\frac{1}{2}G\left(Y_{\left[[b,c]\right]},Y_{a}\right)-\frac{1}{2}G\left(Y_{\left[[b,a]\right]},Y_{c}\right)+\frac{1}{2}G\left(X_{b},X_{\left[[a,c]\right]}\right). \end{array}$$
Therefore, after exploiting Equation (Equation 61) to rewrite the scalar products in the last line of Equation (Equation 59), and after a simple manipulation involving the Jacobi identity for the Lie product [[,]] (see [53]), we obtain
(62)$$\begin{array}{cc} G\left(R_{G}\left(Y_{a},Y_{b}\right)Y_{c},Y_{d}\right) & \;=\;\frac{1}{4}G\left(X_{\left[[b,\{a,c\}]\right]},Y_{d}\right)-\frac{1}{4}G\left(X_{\left[[a,\{b,c\}]\right]},Y_{d}\right)-\frac{1}{4}G\left(Y_{\left\{b,\{a,c\}\right\}},Y_{d}\right)+\\ & \;\;\;\;+\frac{1}{4}G\left(X_{\left[[b,\{a,c\}]\right]},Y_{d}\right)+\frac{1}{4}G\left(Y_{\left[[[[a,b]],c]\right]},Y_{d}\right)+\frac{1}{4}G\left(Y_{\left[[[[a,c]],d]\right]},Y_{b}\right)+\\ & \;\;\;\;+\frac{1}{2}G\left(Y_{\left[[[[a,b]],d]\right]},Y_{c}\right)-\frac{1}{4}G\left(Y_{\left[[[[b,c]],d]\right]},Y_{a}\right)+\\ & \;\;\;\;+\frac{1}{2}G\left(X_{\left[[a,b]\right]},X_{\left[[c,d]\right]}\right)+\frac{1}{4}G\left(X_{\left[[a,c]\right]},X_{\left[[b,d]\right]}\right)-\frac{1}{4}G\left(X_{\left[[b,c]\right]},X_{\left[[a,d]\right]}\right) \end{array}$$
Now, a tedious but straightforward computation based on the expansion of all the Jordan and Lie products according to Equation (Equation 1) shows that
(63)$$\begin{array}{cc} G(R_{G}\left(Y_{a},Y_{b}\right)Y_{c},Y_{d}) & \;=\;\frac{1}{4}G\left(Y_{\left[[a,d]\right]},Y_{\left[[b,c]\right]}\right)-\frac{1}{4}G\left(Y_{\left[[a,c]\right]},Y_{\left[[b,d]\right]}\right)-\frac{1}{2}G\left(Y_{\left[[a,b]\right]},Y_{\left[[c,d]\right]}\right)+\\ & \;\;\;\;+\frac{1}{2}G\left(X_{\left[[a,b]\right]},X_{\left[[c,d]\right]}\right)+\frac{1}{4}G\left(X_{\left[[a,c]\right]},X_{\left[[b,d]\right]}\right)-\frac{1}{4}G\left(X_{\left[[b,c]\right]},X_{\left[[a,d]\right]}\right) \end{array}$$

Now, the sectional curvature $K_{G}$ of G (see [60] for the definition) reads
(64)$$\begin{array}{cc} K_{G}\left(Y_{a},Y_{b}\right) & \;=\;\frac{1}{N}\;G\left(R_{G}\left(Y_{a},Y_{b}\right)Y_{b},Y_{a}\right)\;, \end{array}$$
where
(65)$$N\;:=\;G\left(Y_{a},Y_{a}\right)\;G\left(Y_{b},Y_{b}\right)-{\left(G\left(Y_{a},Y_{b}\right)\right)}^{2}\;=\;l_{a^{2}}^{+}\;l_{b^{2}}^{+}-{\left(l_{\left\{a,b\right\}}^{+}\right)}^{2}\;,$$
and thus we obtain
(66)$$K_{G}\left(Y_{a},Y_{b}\right)\;=\;\frac{3}{4N}\left(G\left(Y_{\left[[a,b]\right]},Y_{\left[[a,b]\right]}\right)-G\left(X_{\left[[a,b]\right]},X_{\left[[a,b]\right]}\right)\right).$$
Note that, when A is Abelian, both the sectional curvature $K_{G}$ and the Riemann curvature $R_{G}$ identically vanish for every orbit O.

## 4. Riemannian Metrics on Manifolds of States

Since we have the Riemannian metric G on the orbit O⊂P, we may consider the orbit $O_{1}\subset O$ and pull back G to $O_{1}$ by means of the immersion map $i_{1+}$ obtaining the Riemannian metric tensor
(67)$$G_{1}\;:=\;i_{1+}^{*}\;G\;.$$
In the following, we will see that this metric tensor determined by the Jordan product “becomes” the Fisher–Rao metric tensor, the Fubini–Study metric tensor, or the Bures–Helstrom metric tensor when we make suitable choices for the explicit form of A and O.

Recalling the definition of the vector fields $Y_{a}$ and $X_{a}$ given in Section 2 below Equation (Equation 23), we immediately obtain
(68)$$\begin{array}{cc} G_{1}\left(Y_{a},Y_{b}\right) & \;=\;i_{1+}^{*}\left(G({\tilde{Y}}_{a},{\tilde{Y}}_{b})\right)\;=\;e_{\left\{a,b\right\}}-e_{a}\;e_{b}\\ G_{1}\left(Y_{a},\;X_{b}\right) & \;=\;i_{1+}^{*}\left(G({\tilde{Y}}_{a},X_{b})\right)\;=\;e_{\left[[b,a]\right]}\\ \left(G_{1}\left(X_{a},X_{b}\right)\right)\left(\rho \right) & \;=\;\left(i_{1+}^{*}\left(G(X_{a},X_{b})\right)\right)\left(\rho \right)\;=\;e_{\left[\left[a,B_{\rho }^{b}\right]\right]}\left(\rho \right)\;=\;\rho \left([\left[a,B_{\rho }^{b}\right]]\right)\;. \end{array}$$
The set ${\left\{Y_{b}\left(\rho \right)\right\}}_{b\in A_{sa}}$ is an overcomplete basis for $T_{\rho }O_{1}$ for every $\rho \in O_{1}$, and from the first line of Equation (Equation 68) and the second relation in Equation (Equation 30), we have that
(69)$$de_{a}\;=\;G_{1}\left(Y_{a},\;\cdot \right),$$
which means that $Y_{a}$ is the gradient vector field associated with $e_{a}$ by means of the Riemannian metric $G_{1}$, and this explains why we already called them gradient vector fields in Section 2.

By adapting the proof used for G, we may prove that the Hamiltonian vector fields preserve $G_{1}$, that is, we may prove that
(70)$$L_{X_{a}}\;G_{1}\;=\;0$$
for all $a\in A_{sa}$. We therefore conclude that $G_{1}$ is invariant with respect to the action of the unitary group U on $O_{1}$ obtained by restricting Φ to U⊂G.

Now, we will compute the covariant derivative $\nabla^{1}$ associated with $G_{1}$. By applying the Koszul formula (see [60] (thm. 4.3.1)) and proceeding as we did in obtaining Equation (Equation 56), we obtain
(71)$$\nabla_{Y_{a}}^{1}Y_{b}\;=\;\frac{1}{2}\left(Y_{\left\{a,b\right\}}-e_{a}Y_{b}-e_{b}Y_{a}-X_{\left[[a,b]\right]}\right).$$
By definition of $G_{1}$, the canonical immersion $i_{1+}:O_{1}⟶O$ is a Riemannian immersion between $\left(O_{1},G_{1}\right)$ and (O,G). Consequently, recalling that $Y_{a}$ is $i_{1+}$-related with $\tilde{Y_{a}}$, we have (see [63] (thm. 1.72))
(72)$$\tilde{\nabla_{Y_{a}}^{1}Y_{b}}\;=\;\nabla_{\tilde{Y_{a}}}\tilde{Y_{b}}+\Pi \left(\tilde{Y_{a}},\tilde{Y_{b}}\right),$$
where $\tilde{\nabla_{Y_{a}}Y_{b}}$ is a vector field on O which is $i_{1+}$-related with $\nabla_{Y_{a}}Y_{b}$, and Π is the second fundamental form of $O_{1}$ in O. From Equations (Equation 23) and (Equation 71), it immediately follows that
(73)$$\tilde{\nabla_{Y_{a}}^{1}Y_{b}}\;=\;\frac{1}{2}\left({\tilde{Y}}_{\left\{a,b\right\}}-l_{a}^{+}\tilde{Y_{b}}-l_{b}^{+}\tilde{Y_{a}}-X_{\left[[a,b]\right]}\right),$$
while, from Equations (Equation 23) and (Equation 56), it immediately follows that
(74)$$\nabla_{\tilde{Y_{a}}}\tilde{Y_{b}}\;=\;\frac{1}{2}\left({\tilde{Y}}_{\left\{a,b\right\}}-l_{a}^{+}\tilde{Y_{b}}-l_{b}^{+}\tilde{Y_{a}}-\left(l_{\left\{a,b\right\}}^{+}-l_{a}^{+}\;l_{b}^{+}\right)Y_{I}-X_{\left[[a,b]\right]}\right),$$
so that
(75)$$\Pi \left(\tilde{Y_{a}},\tilde{Y_{b}}\right)\;=\;\tilde{\nabla_{Y_{a}}^{1}Y_{b}}-\nabla_{\tilde{Y_{a}}}\tilde{Y_{b}}\;=\;\frac{1}{2}\left(l_{\left\{a,b\right\}}^{+}-l_{a}^{+}\;l_{b}^{+}\right)Y_{I}.$$
This means that $O_{1}$ is not a totally geodesic submanifold of O. Then, concerning the Riemann curvature tensor $R_{G_{1}}$ associated with $G_{1}$, we have the standard formula
$$G_{1}\left(R_{G_{1}}\left(Y_{a},Y_{b}\right)Y_{c},Y_{d}\right)=i_{1+}^{*}\left(G\left(R_{G}\left(\tilde{Y_{a}},\tilde{Y_{b}}\right)\tilde{Y_{c}},\tilde{Y_{d}}\right)+G\left(\Pi \left(\tilde{Y_{a}},\tilde{Y_{d}}\right),\Pi \left(\tilde{Y_{b}},\tilde{Y_{c}}\right)\right)-G\left(\Pi \left(\tilde{Y_{a}},\tilde{Y_{c}}\right),\Pi \left(\tilde{Y_{b}},\tilde{Y_{d}}\right)\right)\right),$$
which becomes
(76)$$\begin{array}{cc} G_{1}\left(R_{G_{1}}\left(Y_{a},Y_{b}\right)Y_{c},Y_{d}\right) & \;=\;i_{1+}^{*}\left(G\left(R_{G}\left(Y_{a},Y_{b}\right)Y_{c},Y_{d}\right)\right)+\frac{N_{1}^{abcd}}{4}, \end{array}$$
where
(77)$$N_{1}^{abcd}\;=\;G_{1}\left(Y_{a},Y_{d}\right)\;G_{1}\left(Y_{b},Y_{c}\right)-G_{1}\left(Y_{a},Y_{c}\right)\;G_{1}\left(Y_{b},Y_{d}\right)\;.$$
From this, it is immediate to conclude that, unlike what happens for positive linear functionals, the Riemann curvature tensor of $O_{1}$ does not vanish when A is Abelian. On the other hand, setting $N_{1}\;:=\;N_{1}^{abba}$, the sectional curvature $K_{G_{1}}$ of $G_{1}$ is easily computed to be
(78)$$\begin{array}{cc} K_{G_{1}}\left(Y_{a},Y_{b}\right) & \;=\;\frac{G_{1}\left(R_{G_{1}}\left(Y_{a},Y_{b}\right)Y_{b},Y_{a}\right)}{N_{1}}\;=\;\\ & \;=\;\frac{1}{4}+\frac{3}{4N_{1}}\left(G_{1}\left(Y_{\left[[a,b]\right]},Y_{\left[[a,b]\right]}\right)+{\left(e_{\left[[a,b]\right]}\right)}^{2}-G_{1}\left(X_{\left[[a,b]\right]},X_{\left[[a,b]\right]}\right)\right)\;, \end{array}$$
and we see that the sectional curvature of every $O_{1}$ is constant and equal to $\frac{1}{4}$ when A is Abelian.

## 5. The Fisher–Rao Metric Tensor

Here, we will study the case in which A is Abelian, and provide a link between the Riemannian metric tensor $G_{1}$ and the Fisher–Rao metric tensor $G_{FR}$. First of all, we may consider the Abelian $C^{*}$-algebra $A=C^{n}$ (with n<∞) endowed with component-wise algebraic operations without loss of generality because, up to isomorphism, this is the unique finite-dimensional, Abelian $C^{*}$-algebra. Then, we take the canonical basis ${\left\{e^{j}\right\}}_{j=1,...,n}$ in $C^{n}$, so that every a∈A can be written as
(79)$$a=a_{j}\;e^{j}$$
with $a_{j}\in C$ for all j=1,...,n. In particular, a is a self-adjoint element if and only if $a_{j}$ is real for all j=1,...,n. By considering the dual basis ${\left\{e_{j}\right\}}_{j=1,...,n}$ of ${\left\{e^{j}\right\}}_{j=1,...,n}$, we introduce its associated Cartesian coordinate system ${\left\{p^{j}\right\}}_{j=1,...,n}$ on V, and it is immediate to check that V may be identified with $R^{n}$, the cone P may be identified with the positive orthant $P^{n}$ in $R^{n}$ (without the zero), and the space of states S may be identified with the standard *n*-simplex ${∆}^{n}$ in $R^{n}$.

Concerning the orbits of G, we first fix a reference element $\omega_{*}\in P$ such that $p^{j}\left(\omega_{*}\right)\neq 0$ only for a subset $J_{*}\subseteq 2^{\left\{1,...,n\right\}}$, and then note that the orbit O of G through $\omega_{*}\in P$ is given by all those positive linear functionals ω such that $p^{j}\left(\omega \right)\neq 0$ if and only if $j\in J_{*}$. In particular, every orbit O may be identified with the open interior of the positive cone in $R^{m}$ with $m=\mathrm{card}(J_{*})$ (cardinality of $J_{*}$).

Similarly, if we fix a reference state $\rho_{*}\in S$ such that $p^{j}\left(\rho_{*}\right)\neq 0$ only for a subset $J_{*}\subseteq 2^{\left\{1,...,n\right\}}$ (i.e., a probability vector supported on $J_{*}$), we have that the orbit $O_{1}$ of G through $\rho_{*}\in S$ is given by all those states ρ such that $p^{j}\left(\rho \right)\neq 0$ if and only if $j\in J_{*}$. In particular, every orbit $O_{1}$ may be identified with the open interior of the *m*-simplex in $R^{m}$ with $m=\mathrm{card}(J_{*})$.

The linear function $l_{a}$ associated with $a\in A_{sa}$ reads
(80)$$l_{a}\;=\;a_{j}\;p^{j}\;,$$
and we have that
(81)$$l_{\left[[a,b]\right]}\;=\;0\;\;\;\forall \;\;a,b\;\;\in A_{sa}$$
because A is Abelian, and
(82)$$l_{\left\{a,b\right\}}\;=\;\sum_{j=1}^{n}\;a_{j}\;b_{j}\;p^{j}$$
because A is Abelian and its product operation is defined component-wise. This means that the tensor field Λ vanishes, while the tensor field R may be written as
(83)$$R\;=\;\sum_{j=1}^{n}\;p^{j}\;\frac{\partial }{\partial p^{j}}\;⊗\;\frac{\partial }{\partial p^{j}}.$$
It is then clear that the “inverse” G of R on the orbit O may be written as
(84)$$G\;=\;\sum_{j\in J_{*}}\;\frac{1}{p^{j}}\;dp^{j}\;⊗\;dp^{j},$$
and the pullback tensor
(85)$$G_{1}\;=\;i_{1+}^{*}\;G,$$
on $O_{1}$ coincides with the canonical Fisher–Rao metric tensor $G_{FR}$ (see [5] (sec. 2.2)) on $O_{1}$ when the latter is considered as the open interior of the *m*-simplex in $R^{m}$ with $m=\mathrm{card}(J_{*})$.

Quite provocatively, we may say that the Fisher–Rao metric tensor is a built-in feature of the $C^{*}$-algebraic approach to classical probability theory on finite sample spaces, and that, in this context, the natural object to start with is the ***contravariant*** tensor field R from which the Fisher–Rao metric tensor (a ***covariant*** tensor field) may be recovered as explained above.

## 6. From the Gelfand-Naimark-Segal Construction to Riemannian Geometries

In this section, we will take inspiration from the reduction procedure adopted to define the Fubini–Study metric tensor on the manifold of pure quantum states (see [64,65]), as well as from Uhlmann’s purification procedure adopted to define the Bures–Helstrom metric tensor on the manifold of faithful quantum states (see [44,45,46,47,66,67]), to give a more appealing geometrical picture of the metric tensors on the manifolds of states on A introduced in the previous section. Essentially, we will build a geometrical picture that is somehow dual to the one presented before. Indeed, in Section 4, the Riemannian manifold $\left(O_{1},G_{1}\right)$ was thought of as the ***source*** of a Riemannian immersion into the Riemannian manifold (O,G), while here, the Riemannian manifold $\left(O_{1},G_{1}\right)$ will be shown to be the ***target*** of a Riemannian submersion from an open submanifold of a suitably big sphere.

In order to develop our ideas, we need to briefly recall some aspects of the geometry of a complex Hilbert space H as a real Kähler manifold (see [64,68,69]). First of all, every complex Hilbert space H may always be thought of as a real, smooth Hilbert manifold (much of what we will say applies also to infinite-dimensional Hilbert spaces, but we will confine the discussion to the finite-dimensional case). Indeed, we may always consider the realification $H_{R}$ as the model space. The realification $H_{R}$ is obtained by restricting linear combinations of elements in H to have only real coefficients, and by defining a real Hilbert product as the real part of the complex Hilbert product on H. In the following, we will write H to denote both the complex Hilbert space and the real, smooth Hilbert manifold modeled on $H_{R}$. This should not induce confusion since the context will always clarify which is the mathematical aspect of H we are referring to. The tangent space $T_{\psi }H$ at ψ∈H may be identified with H itself because H is a vector manifold. Consequently, we may set
(86)$$\Omega \left(X_{\psi },\;Y_{\psi }\right)\;:=\;\frac{2}{ı}\left(\langle X_{\psi },\;Y_{\psi }\rangle -\langle Y_{\psi },\;X_{\psi }\rangle \right),$$
and
(87)$$E\left(X_{\psi },\;Y_{\psi }\right)\;:=\;2\left(\langle X_{\psi },\;Y_{\psi }\rangle +\langle Y_{\psi },\;X_{\psi }\rangle \right),$$
so that a direct computation shows that Ω is a symplectic form and that E is a Riemannian metric tensor.

Now, let us fix a state ρ. We denote by H the GNS Hilbert space associated with ρ, by $\psi_{I}$ the GNS vector associated with ρ, and with r the GNS representation of A in B(H). Every vector $\psi_{a}\in H$ gives rise to a positive linear functional on A given by
(88)$$\omega_{\psi_{a}}\left(c\right)\;:=\;\langle \psi_{a}\left|r\left(c\right)\right|\psi_{a}\rangle \;,$$
in particular, if the vector is normalized, the associated positive linear functional is a state.

For every $\psi_{a}\in H_{0}=H-\left\{0\right\}$ we set
(89)$$\hat{\psi_{a}}\;=\;\frac{\psi_{a}}{\sqrt{\langle \psi_{a}|\psi_{a}\rangle }}$$
so that $\hat{\psi_{a}}$ is automatically on the unit sphere $S_{1}$ in H, and every element in $S_{1}$ may be written as $\hat{\psi_{a}}$ for some $\psi_{a}$ with $a\notin N_{\rho }$. Note that the condition $\langle \psi_{a}|\psi_{a}\rangle =\rho \left(a^{†}a\right)=0$ means that a is in the ideal $N_{\rho }$, which means that $\psi_{a}$ is the null vector in H. Then, the map $\pi :S_{1}⟶S$ given by
(90)$$\hat{\psi_{a}}\;\mapsto \;\pi \left(\hat{\psi_{a}}\right)\;:=\;\rho_{\hat{\psi_{a}}}$$
is easily seen to be continuous with respect to the topology underlying the standard differential structure of $S_{1}$ and the weak* topology on S (which in the finite-dimensional case coincides with the norm topology). Furthermore, the image of $S_{1}$ through π consists of all those states $\rho_{\hat{\psi_{a}}}$ acting as
(91)$$\rho_{\hat{\psi_{a}}}\left(c\right)\;=\;\frac{\rho (a^{†}\;c\;a)}{\rho (a^{†}a)}.$$
In particular, from this last equation we immediately see that the smooth homogeneous space $O_{1}$ containing ρ is in the image of $S_{1}$ through π.

Now, in the finite-dimensional case, the Hilbert space H is just the quotient $A/N_{\rho }$, where $N_{\rho }$ is the Gel’fand ideal associated with ρ. There is a natural projection map pr:A⟶H, and this projection map is an open map because $H=A/N_{\rho }$ is the quotient by a group action. Consequently, the space
(92)$$H\left(G\right)\;:=\;\left\{\psi \in H\;:\;\;\exists \;g\in G\;|\;\;\psi \;=\;\mathrm{pr}(g)\right\}$$
is open in H because it is the image pr(G) of the open set G⊂A, and if we set
(93)$$S_{1}\left(G\right)\;:=\;H\left(G\right)\;\cap \;S_{1},$$
we immediately conclude that $S_{1}\left(G\right)$ is an open submanifold of the unit sphere $S_{1}$, and that, essentially by definition, every $\psi \in S_{1}\left(G\right)$ is such that there exists an invertible element g∈G such that $\psi \;=\mathrm{pr}\left(g\right)\;=\;\hat{\psi_{g}}$. This means that the image $\pi (S_{1}\left(G\right))$ in S through the map π defined in Equation (Equation 90) coincides with the orbit $O_{1}\subset S$ and vice versa. This is analogous to the correspondence between the positive octant of the sphere and the unit simplex, as exploited in [9].

We note that there is a left action β of G on the sphere $S_{1}$ given by
(94)$$\left(g,\hat{\psi_{c}}\right)\;\mapsto \;\beta \left(g,\hat{\psi_{c}}\right)\;=\;\frac{\left(r\left(g\right)\right)\left(\psi_{c}\right)}{\sqrt{\langle \psi_{c}|r\left(g^{†}\;g\right)|\psi_{c}\rangle }}\;\equiv \;\hat{\left(r\left(g\right)\right)\left(\psi_{c}\right)}\;.$$
This action is smooth, and its fundamental vector fields $\Psi_{ab}$ are easily seen to be
(95)$$\Psi_{ab}\left(\hat{\psi_{c}}\right)\;=\;\frac{1}{2}\left(r(a)+ı\;r(b)\right)\;\left(\hat{\psi_{c}}\right)-\frac{1}{2}\langle \hat{\psi_{c}}|r\left(a\right)|\hat{\psi_{c}}\rangle \;\hat{\psi_{c}}\;,$$
where we implicitly assumed that $g=e^{\frac{1}{2}\left(a+ıb\right)}$ with $a,b\in A_{sa}$. Essentially by definition, this action preserves the open submanifold $S_{1}\left(G\right)$ of the unit sphere, and it is actually transitive on it as can be checked by direct inspection. Consequently, the vector fields $\Psi_{ab}$ provide an overcomplete basis for the tangent space at each $\hat{\psi_{g}}\in S_{1}\left(G\right)$.

**Proposition** **3.**
*The map $\pi \;:\;\;S_{1}\left(G\right)\;⟶\;O_{1}$ obtained by restricting the map π of Equation (Equation 90) is a submersion.*


**Proof.** To prove the proposition, we will show that
(96)$$T_{\hat{\psi_{g}}}\pi \left(\Psi_{ab}\left(\hat{\psi_{g}}\right)\right)\;=\;Y_{ab}\left(\pi \left(\hat{\psi_{g}}\right)\right),$$
where $Y_{ab}$ is the fundamental vector field of the (transitive) action Φ of G on $O_{1}$ (see Equation (Equation 24)), which proves that $T_{\hat{\psi_{g}}}\pi$ is surjective for every $\hat{\psi_{g}}\in S_{1}\left(G\right)$. The proof of Equation (Equation 96) is obtained by noting that
(97)$$\pi^{*}e_{c}\left(\hat{\psi_{g}}\right)\;=\;e_{c}\left(\pi \left(\hat{\psi_{g}}\right)\right)\;=\;\rho_{\hat{\psi_{g}}}\left(c\right)\;=\;\frac{\langle \psi_{g}|r\left(c\right)|\psi_{g}\rangle }{\langle \psi_{g}|\psi_{g}\rangle }\;=\;\langle \hat{\psi_{g}}|r\left(c\right)|\hat{\psi_{g}}\rangle ,$$
and then directly computing
(98)$$\begin{array}{cc} \frac{d}{dt}\;{\left(\pi^{*}e_{c}\left({\mathrm{Fl}}_{t}^{\Psi_{ab}}\left(\hat{\psi_{g}}\right)\right)\right)}_{t=0} & \;=\;\frac{d}{dt}\;{\left(\frac{\langle \psi_{g}|r\left(e^{\frac{t}{2}\left(a-ıb\right)}\right)\;r\left(c\right)\;r\left(e^{\frac{t}{2}\left(a+ıb\right)}\right)|\psi_{g}\rangle }{\langle \psi_{g}|r\left(e^{\frac{t}{2}\left(a-ıb\right)}\right)\;r\left(e^{\frac{t}{2}\left(a+ıb\right)}\right)|\psi_{g}\rangle }\right)}_{t=0}\;=\;\\ & \;=\;\frac{\langle \psi_{g}|r\left(\{a,c\}+[[b,c]]\right)|\psi_{g}\rangle }{\langle \psi_{g}|\psi_{g}\rangle }-\frac{\langle \psi_{g}|r\left(c\right)|\psi_{g}\rangle }{\langle \psi_{g}|\psi_{g}\rangle }\;\frac{\langle \psi_{g}|r\left(a\right)|\psi_{g}\rangle }{\langle \psi_{g}|\psi_{g}\rangle }\;=\;\\ & \;=\;e_{\left\{a,c\right\}}\left(\pi \left(\hat{\psi_{g}}\right)\right)+e_{\left[[b,c]\right]}\left(\pi \left(\hat{\psi_{g}}\right)\right)-e_{a}\left(\pi \left(\hat{\psi_{g}}\right)\right)\;e_{c}\left(\pi \left(\hat{\psi_{g}}\right)\right)\;. \end{array}$$ □

**Remark** **2.**
*From this proposition, we conclude that every $\Psi_{ab}$ is π-related with the fundamental vector field $Y_{ab}=Y_{a}+X_{b}$ of the action Φ of G on $O_{1}$ given in Equation (Equation 27), and thus the action of G on $O_{1}$ may be seen as the projected shadow of the action of G on $S_{1}\left(G\right)$. In particular, the same is true for the action of the unitary group U on $O_{1}$. Furthermore, the validity of Equation (Equation 96) implies that the kernel of $T_{\hat{\psi_{g}}}\pi$ coincides with the isotropy algebra $g_{\pi (\hat{\psi_{g}})}^{\Phi }$ of the action Φ at $\pi (\hat{\psi_{g}})$. This is consistent with the fact that the equality*
(99)$$\pi \left(\hat{\psi_{g}}\right)\;=\;\pi \left(\hat{\psi_{h}}\right)$$
*is equivalent to*
(100)h=kg
*with k in the isotropy subgroup $G_{\pi (\hat{\psi_{g}})}$ of $\pi \left(\hat{\psi_{g}}\right)\in O_{1}\subset S$ with respect to Φ, as can easily be checked.*


On the unit sphere $S_{1}$ there is the action of another relevant Lie group, namely, the Lie group $U^{\prime }\subset B\left(H\right)$, which consists of unitary elements in the commutant $A^{\prime }$ of r(A) in B(H). The action of $U^{\prime }$ on $S_{1}$ is just the restriction of its natural action on H, and this action is proper because both $U^{\prime }$ and $S_{1}$ are compact (in the finite-dimensional case). The fundamental vector fields of this action will be denoted by $\Xi_{b}$, where $B\in A^{\prime }\subset B\left(H\right)$ is skew-adjoint (i.e., $B^{†}=-B$), and it is possible to prove that every $\Xi_{b}$ commutes with every $\Psi_{ab}$. Indeed, the flow ${\mathrm{Fl}}_{t}^{\Psi_{ab}}$ of $\Psi_{ab}$ on $\hat{\psi_{c}}$ is given by (see [59] for the definition of the flow of a vector field)
(101)$${\mathrm{Fl}}_{t}^{\Psi_{ab}}\left(\hat{\psi_{c}}\right)\;=\;\frac{\left(r\left(g_{t}\right)\right)\left(\psi_{c}\right)}{\sqrt{\langle \psi_{c}|r\left(g_{t}^{†}\;g_{t}\right)|\psi_{c}\rangle }}\;$$
with $g_{t}=e^{\frac{t}{2}\left(a+ıb\right)}$ (see Equation (Equation 94)), while the flow ${\mathrm{Fl}}_{t}^{\Xi_{b}}$ of $\Xi_{b}$ on $\hat{\psi_{c}}$, by definition, is just
(102)$${\mathrm{Fl}}_{t}^{\Xi_{b}}\left(\hat{\psi_{c}}\right)\;=\;U_{t}^{\prime }\left(\hat{\psi_{c}}\right)$$
with $U_{t}^{\prime }=e^{tB}$, and, since $U_{t}^{\prime }\in A^{\prime }$, it immediately follows that the flows of these vector fields commute, i.e., the vector fields themselves commute.

**Proposition** **4.**
*The group $U^{\prime }$ acts freely and properly on $S_{1}\left(G\right)$, and there is a diffeomorphism between the quotient space $M=S_{1}\left(G\right)/U^{\prime }$ endowed with the canonical smooth structure coming from the free and proper action and the smooth manifold $O_{1}$ endowed with the smooth structure recalled in Section 2.*


**Proof.** First of all, a direct computation shows that
(103)$$\pi \left(U^{\prime }\left(\hat{\psi_{g}}\right)\right)\;=\;\pi \left(\hat{\psi_{g}}\right)$$
for every $U^{\prime }\in U^{\prime }$ and every $\hat{\psi_{g}}\in S_{1}\left(G\right)$, and thus the action of $U^{\prime }$ preserves $S_{1}\left(G\right)$ because $S_{1}\left(G\right)$ is the preimage of $O_{1}$ through π. Then, we note that the action of $U^{\prime }$ in $S_{1}\left(G\right)$ is proper for the group being compact. To show that it is free, note that, by construction of the GNS representation, the vector $\psi_{I}$ is cyclic for r(A), and it is separating for $A^{\prime }$ because of a standard result (see [49] (prop. 2.5.3)). Consequently, if $a^{\prime }\in A^{\prime }$ is such that
(104)$$a^{\prime }\left(\psi_{I}\right)\;=\;0,$$
then $a^{\prime }$ is the zero element in B(H). This means that the isotropy group $U_{\psi_{I}}^{\prime }$ of $\psi_{I}$ with respect to the action of $U^{\prime }\subset A^{\prime }$ is the trivial group consisting only of the identity operator on H. Furthermore, the isotropy group $U_{\psi_{g}}^{\prime }$ of every $\psi_{g}\in S_{1}\left(G\right)$ is the trivial subgroup. Indeed, every $\psi_{g}\in S_{1}\left(G\right)$ is cyclic for r(A) because
(105)$$\psi_{c}\;=\;\left(r\left(cg^{-1}\right)\right)\left(\psi_{g}\right)\;\;\;\;\forall \;\psi_{c}\in H,$$
and thus $\psi_{g}$ is separating for $A^{\prime }$ (see [49] (prop. 2.5.3)) and we may proceed as before. From this, we have that the quotient space $M=S_{1}\left(G\right)/U^{\prime }$ is a smooth manifold.The group G acts on *M* by means of the projection of the action β on $S_{1}\left(G\right)$ introduced before because the fundamental vector fields $\Psi_{ab}$ generating β commute with the fundamental vector fields $\Xi_{b}$ generating the action of $U^{\prime }$ giving rise to *M*. Furthermore, this action is transitive on *M* because β is transitive on $S_{1}\left(G\right)$. We denote this action by $\tilde{\Phi }$, and we have
(106)$$\tilde{\Phi_{h}}\left(\left[\hat{\psi_{g}}\right]\right)\;=\;\left[\beta_{h}\left(\hat{\psi_{g}}\right)\right].$$
From this it follows that h is in the isotropy group of $\left[\hat{\psi_{g}}\right]$ with respect to $\tilde{\Phi }$ if and only if (recall that the isotropy subgroup of $\left[\hat{\psi_{g}}\right]$ with respect to $\tilde{\Phi }$ is the set of all group elements in $U^{\prime }$ leaving $\left[\hat{\psi_{g}}\right]$ unaltered)
(107)$$\left[\beta_{h}\left(\hat{\psi_{g}}\right)\right]\;=\;\left[\hat{\psi_{g}}\right],$$
that is, if and only if there exists a unitary element $U_{h}^{\prime }$ such that
(108)$$\beta_{h}\left(\hat{\psi_{g}}\right)\;=\;U_{h}^{\prime }\left(\hat{\psi_{g}}\right).$$
Clearly, this means that
(109)$$\pi \left(\beta_{h}\left(\hat{\psi_{g}}\right)\right)\;=\;\pi \left(U_{h}^{\prime }\left(\hat{\psi_{g}}\right)\right)\;=\;\pi \left(\hat{\psi_{g}}\right),$$
and thus h is in the isotropy subgroup $G_{\pi (\hat{\psi_{g}})}$ of $\pi \left(\hat{\psi_{g}}\right)\in O_{1}\subset S$ with respect to Φ (see Remark 2). Therefore, $[\hat{\psi_{g}}]\in M$ and $\pi \left(\hat{\psi_{g}}\right)\in O_{1}$ have the same isotropy group for every $\hat{\psi_{g}}\in S_{1}\left(G\right)$, and this implies that there is a diffeomorphism between *M* endowed with the differential structure and the G-action coming from the quotient procedure on $S_{1}\left(G\right)$ and the manifold $O_{1}\subset S$ endowed with the smooth structure and the G-action recalled in Section 2.Note, however, that this diffeomorphism does not extend to the boundaries where it is only a homeomorphism. □

At this point, we consider the metric tensor *g* on $S_{1}\left(G\right)$ which is the pullback of the Euclidean tensor E on H by means of the canonical immersion of $S_{1}\left(G\right)$ into H given by the identification map, and we will prove that the projection map $\pi :S_{1}\left(G\right)⟶O_{1}$ is a Riemannian submersion between $\left(S_{1}\left(G\right),g\right)$ and $\left(O_{1},G_{1}\right)$. First of all, we note that the vector fields $\Xi_{b}$ generating the action of $U^{\prime }$ on $S_{1}\left(G\right)$ span the kernel of π because of Proposition 4. Then, a direct computation shows that
(110)$$g(\Psi_{a0},\Xi_{b})\;=\;0$$
for every $a\in A_{sa}$ and every B in the Lie algebra of $U^{\prime }$. Indeed, we have
(111)$$g_{\hat{\psi_{g}}}\left(\Psi_{a0}\left(\hat{\psi_{g}}\right),\Xi_{b}\left(\hat{\psi_{g}}\right)\right)\;=\;2\left(\langle \hat{\psi_{g}}\left|r\left(a\right)B\right|\hat{\psi_{g}}\rangle +\langle \hat{\psi_{g}}|B^{†}\;r\left(a\right)|\hat{\psi_{g}}\rangle \right)\;=\;0$$
because B is skew-adjoint and commutes with r(a). This means that the linear span of the $\Psi_{a0}$’s is in the orthogonal complement of the vertical distribution. However, π being a submersion, we have
(112)$$\dim\left(S_{1}\left(G\right)\right)\;=\;\dim\left(O_{1}\right)+\dim\left(U^{\prime }\right),$$
and since the $\Psi_{a0}$’s are π-related with the gradient vector fields $Y_{a}$ (see Remark 2), and these vector fields provide an overcomplete basis of tangent vectors at each point in $O_{1}$, we conclude that the linear span of the $\Psi_{a0}$’s generates the whole orthogonal complement of the vertical distribution at each point in $S_{1}\left(G\right)$. What is left to prove is that
(113)$$g\left(\Psi_{a0},\;\Psi_{b0}\right)\;=\;G_{1}\left(Y_{a},\;Y_{b}\right)$$
for all $a,b\in A_{sa}$. For this purpose, recalling first Equation (Equation 95), then Equation (Equation 97), and then Equation (Equation 68), we have
(114)$$\begin{array}{cc} g_{\hat{\psi_{g}}}\left(\Psi_{a0}\left(\hat{\psi_{g}}\right),\Psi_{b0}\left(\hat{\psi_{g}}\right)\right) & \;=\;\frac{1}{2}\left(\langle \hat{\psi_{g}}|r\left(a\right)r\left(b\right)+r\left(b\right)r\left(a\right)|\hat{\psi_{g}}\rangle \right)-\langle \hat{\psi_{g}}\left|r\left(a\right)\right|\hat{\psi_{g}}\rangle \langle \hat{\psi_{g}}\left|r\left(b\right)\right|\hat{\psi_{g}}\rangle \;=\;\\ & \;=\;\rho_{\hat{\psi_{g}}}\left(\{a,\;b\}\right)-\rho_{\hat{\psi_{g}}}\left(a\right)\;\rho_{\hat{\psi_{g}}}\left(b\right)\;=\;\\ & \;=\;e_{\left\{a,b\right\}}\left(\pi \left(\hat{\psi_{g}}\right)\right)-e_{a}\left(\pi \left(\hat{\psi_{g}}\right)\right)\;e_{b}\left(\pi \left(\hat{\psi_{g}}\right)\right)\;=\;\\ & \;=\;{\left(G_{1}\right)}_{\pi (\hat{\psi_{g}})}\;\left(Y_{a}\left(\pi \left(\hat{\psi_{g}}\right)\right),\;Y_{b}\left(\pi \left(\hat{\psi_{g}}\right)\right)\right), \end{array}$$
which proves the validity of Equation (Equation 113), which implies that π is a Riemannian submersion between $\left(S_{1}\left(G\right),g\right)$ and $\left(O_{1},G_{1}\right)$ as claimed.

The fact that π is a Riemannian submersion implies that every geodesic of $G_{1}$ on $O_{1}$ may be written as the projection of a geodesic of *g* on $S_{1}\left(G\right)$ having initial tangent vector in the horizontal distribution. Therefore, since $S_{1}\left(G\right)$ is an open submanifold of the unit sphere in H, and since *g* is the restriction of four times the round metric on $S_{1}$, we have that the explicit expression of the geodesic γ of *g* on $S_{1}\left(G\right)$ starting at $\hat{\psi_{g}}$ with initial (constant) tangent vector ϕ≠0 is given by
(115)$$\gamma \left(t\right)\;=\;\cos\left(|\phi |t\right)\;\hat{\psi_{g}}+\sin\left(|\phi |t\right)\frac{\phi }{\left|\phi \right|},$$
where ${\left|\phi \right|}^{2}=\langle \phi |\phi \rangle$. The tangent vector ϕ is horizontal if and only if there exists $a\in A_{sa}$ such that $\phi =\Psi_{a0}\left(\hat{\psi_{g}}\right)$, and it is different from the null vector if and only if a is not in the Gel’fand ideal generated by $\rho_{\hat{\psi_{g}}}=\pi \left(\hat{\psi_{g}}\right)$. Therefore, assuming a not to be in the Gel’fand ideal generated by $\rho_{\hat{\psi_{g}}}=\pi \left(\hat{\psi_{g}}\right)$, a direct computation shows that the geodesic σ starting at $\rho_{g}\in O_{1}$ with initial tangent vector $Y_{a}\left(\rho_{g}\right)$ reads
(116)$$\sigma \left(t\right)\;=\;\cos^{2}\left(N_{g,a}\;t\right)\;\rho_{g}+\frac{\sin^{2}\left(N_{g,a}\;t\right)}{N_{g,a}^{2}}\;\rho_{a_{g}}+\frac{\sin(2N_{g,a}\;t)}{2\;N_{g,a}}\;\left\{\rho_{a_{g}}\right\}\;,$$
where we have set
(117)$$\begin{array}{cc} \rho_{g} & \;:=\;\rho_{\hat{\psi_{g}}},\\ a_{g} & \;:=\;a-\rho_{g}\left(a\right)\;I,\\ \rho_{a_{g}}\left(\cdot \right) & \;:=\;\rho_{g}\left(a_{g}\;\left(\cdot \right)\;a_{g}\right),\\ \left\{\rho_{a_{g}}\right\}\left(\cdot \right) & \;:=\;\rho_{g}\left(\{a_{g},\cdot \}\right)\\ N_{g,a}^{2} & \;:=\;\rho_{g}\left(a^{2}\right)-{\left(\rho_{g}\left(a\right)\right)}^{2}. \end{array}$$
In general, this geodesic “leaves” $O_{1}$ remaining in S, and after some time it returns in $O_{1}$ essentially because the geodesic γ of which σ is the projection is a great circle on a sphere. A case in which $O_{1}$ is geodesically complete is when A is the algebra of linear operators on a finite-dimensional, complex Hilbert space and $O_{1}$ is the (compact) manifold of pure states. As we will see below, in this case $G_{1}$ corresponds to the Fubini–Study metric tensor. Note that, when $A=C^{n}$, Equation (Equation 116) gives an explicit form for the geodesics of the Fisher–Rao metric tensor.

**Remark** **3.**
*Note that the procedure applied here to $S_{1}\left(G\right)$ may be adapted in the obvious way to the open submanifold H(G) of H introduced in Equation (Equation 92). The result is that we obtain a Riemannian submersion between (H(G),E) (where E is the Euclidean metric tensor on H given in Equation (Equation 87)) and (O,G) where O is the orbit of G in the space of positive linear functionals P containing the reference state ρ thought of as an element of P. Accordingly, it is possible to obtain an explicit expression also for the geodesics of G.*


## 7. The Fubini–Study Metric Tensor

We will now explicitly perform the construction presented above in the case where A is the algebra B(H) of linear operators on the finite-dimensional, complex Hilbert space H, and we consider the orbit $O_{1}$ of pure states. In doing this, we will essentially recover the standard construction of the diffeomorphism of the complex projective space CP(H) associated with H with the manifold $O_{1}$ of pure states, and we will see that the Riemannian metric $G_{1}$ on $O_{1}≅\mathrm{CP}\left(H\right)$ is (a multiple of) the Fubini–Study metric tensor.

First of all, let us recall that the space of pure states on B(H) is the space of rank one projectors on H (this is true only in the finite-dimensional case; if H is infinite-dimensional, then the space of rank one projectors on H is the space of ***normal*** pure states), that is, a pure state $\rho_{\psi }$ on B(H) may always be written as
(118)$$\rho_{\psi }\;=\;\frac{\left|\psi \rangle \langle \psi \right|}{\langle \psi |\psi \rangle }$$
for some non-zero vector ψ∈H.

Let us fix a normalized vector ψ∈H, and let us introduce an orthonormal basis ${\left\{e_{j}\right\}}_{j=1,...,\dim(H)}$ in H such that $e_{1}=\psi$. Then, the Gel’fand ideal $N_{\rho_{\psi }}$ of the reference state $\rho_{\psi }$ is easily seen to be given by all those linear operators a on H that can be written as
(119)$$a\;=\;\sum_{k\neq 1}\;a_{jk}\;|e_{j}\rangle \langle e_{k}|\;.$$
From this, we obtain that the GNS Hilbert space associated with $\rho_{\psi }$ can be identified with H itself, and thus the GNS representation of B(H) on the GNS Hilbert space may be identified with B(H) itself. Moreover, the GNS representation is irreducible (as it must be because we are considering the GNS representation associated with a pure state) and its commutant is given by the multiples of the identity operator on H.

In the case we are considering, it is immediate to check that the open submanifold H(G) defined in Equation (Equation 92) coincides with $H_{0}\;=\;H-\left\{0\right\}$, and we conclude that
(120)$$S_{1}\left(G\right)\;=\;H\left(G\right)\cap S_{1}\;=\;H_{0}\cap S_{1}\;=\;S_{1}\;.$$

The unitary group $U^{\prime }$ of the commutant of A=B(H) is just the action of the phase group U(1) consisting of elements of the form $e^{ı\theta }I$, with θ∈R. Therefore, in the case at hand, the quotient space $S_{1}\left(G\right)/U^{\prime }≅O_{1}$ appearing in the general construction presented in the previous section is just $S_{1}/U\left(1\right)$, which is precisely the complex projective space associated with H, and we obtain the well-known diffeomorphism between the manifold $O_{1}$ of pure states on B(H) thought of as rank one projectors with the complex projective space CP(H). Under this isomorphism, the action of the unitary group U=U(H) on $O_{1}$ coincides with the canonical action of the unitary group on the complex projective space, and this is enough to conclude that the pullback to CP(H) of the Riemannian metric $G_{1}$ on $O_{1}$ is a multiple of the Fubini–Study metric tensors. Indeed, we know that $G_{1}$ is invariant with respect to the action of the unitary group on $O_{1}$ (see Section 4), and thus its pullback on CP(H) will be invariant with respect to the canonical action of the unitary group on the complex projective space, and this forces this metric to be a multiple of the Fubini–Study metric tensor, since the latter as a metric of a symmetric space is characterized by that property, see for instance [60].

## 8. The Bures–Helstrom Metric Tensor

In this subsection, we will explore the case where A is again the algebra B(H) of linear operators on the finite-dimensional, complex Hilbert space H, but the orbit $O_{1}$ is the orbit of faithful states. As will be clear, in this case we obtain Uhlmann’s construction of the Bures–Helstrom metric tensor (see [44,45,46,47]). Sometimes, this metric tensor is also called the Bures metric, or the Quantum Fisher Information Matrix.

First of all, recall that the Hilbert space trace Tr(·) gives an isomorphism between A and its dual $A^{*}$. Essentially, every $\xi \in A^{*}$ may be identified with an element in A, denoted again by ξ with an evident abuse of notation, such that
(121)ξ(a)=Tr(ξa)∀a∈A.
In view of the literature on the quantum information of finite-level systems, in the remainder of this subsection, we will always maintain a *bipolar* attitude and think of ξ as either an element of A or of $A^{*}$, hoping that no serious confusion arises. Accordingly, the vector space V is also thought of as the space $A_{sa}$ of self-adjoint elements in A, P is also thought of as the space of positive elements in A, S is also thought of as the space of density operators in A.

Now, we fix the reference state τ to be the maximally mixed state associated with the element $\frac{I}{n}$, where n=dim(H). Since the reference state is faithful, then its Gel’fand ideal contains only the zero element in the algebra A=B(H), and thus the vector space underlying the GNS Hilbert space is B(H) itself. Therefore, the Hilbert product in the GNS Hilbert space $H_{\tau }$ reads
(122)$$\langle \psi_{a}|\psi_{b}\rangle \;=\;\tau \left(a^{†}\;b\right)\;=\;\frac{1}{n}\mathrm{Tr}\left(a^{†}\;b\right),$$
and we conclude that the GNS Hilbert space is essentially the Hilbert space of Hilbert–Schmidt operators on H. Then, we easily obtain that the open submanifold H(G) defined in Equation (Equation 92) coincides with G itself, and thus $S_{1}\left(G\right)=H\left(G\right)\cap S_{1}$ coincides with the set of all the invertible elements in the algebra satisfying the normalization condition
(123)$$\frac{1}{n}\;\mathrm{Tr}\left(g^{†}\;g\right)\;=\;1\;.$$
The GNS representation of B(H) coincides with the left action of B(H) on itself, and its commutant coincides with B(H) acting by means of the right action on itself. Consequently, if ρ is a faithful state on B(H), that is, an invertible, positive operator on H with unit trace, we have that all the vectors $\hat{\psi_{g}}\in S_{1}\left(G\right)$ such that their projections $\pi (\hat{\psi_{g}})$ coincide with ρ may be written as gu, where g∈G satisfies the normalization condition given in Equation (Equation 123), and u is a unitary element in G, that is, a unitary operator on H. Comparing what we have just obtained with the results in [44,45,46,47], it follows that the general construction presented in Section 6 essentially reduces to Uhlmann’s purification procedure used to define the Bures–Helstrom metric tensor. However, we will now give a more explicit proof of the equivalence of $G_{1}$ with the Bures–Helstrom metric tensor, which will also stress the fact that some of the computational difficulties usually associated with the expression of the Bures–Helstrom metric tensor (as explicitly stated for instance in [39] below equation 9.43) may be attributed to a particular realization of the tangent space at each $T_{\rho }O_{1}$, which is not well-suited for the Bures–Helstrom metric tensor.

To better appreciate this instance, we recall that, in the standard approach to the definition of the Bures–Helstrom metric tensor, the manifold $S_{+}$ of faithful states on H is identified with the manifold of invertible density operators, the tangent space $T_{\rho }S_{+}$ is identified with the affine hyperplane $A_{sa}^{0}$ of self-adjoint elements in A=B(H) with vanishing trace. This realization of $T_{\rho }S_{+}$ is clearly different from the one used here in terms of the gradient vector fields, and we will now analyze their relation. The gradient vector fields on V allow to identify a tangent vector at ξ∈V with an element in $A_{sa}$ by means of
(124)$$Y_{a}\left(\xi \right)\;=\;\left\{\xi ,\;a\right\},$$
where ξ is thought of as an element of $A_{sa}$. It is worth noting that, when we consider a state ρ, the tangent vector $Y_{a}\left(\rho \right)$ provides a geometrical version of the Symmetric Logarithmic Derivative at ρ widely used in quantum estimation theory [42,43,70,71,72]. On the other hand, since $V≅A_{sa}$, we may also define a constant vector field associated with every $a\in A_{sa}$ by setting
(125)$$Z_{a}\left(\xi \right)\;=\;a\;.$$
If we restrict our considerations to the open submanifold $P_{+}\subset V$ of faithful positive linear functionals, we may relate gradient vector fields with constant vector fields as follows. First of all, recall that every $\omega \in P_{+}$ is identified with an invertible element in $A_{sa}$, and we may write
(126)$$A_{\omega }\left(a\right)\;:=\;\left\{\omega ,\;a\right\}\;=\;\frac{1}{2}\;\left(L_{\omega }+R_{\omega }\right)\left(a\right),$$
where $L_{\omega }:A_{sa}⟶A_{sa}$ is the linear operator given by $L_{\omega }\left(a\right):=\omega \;a$, and $R_{\omega }:A_{sa}⟶A_{sa}$ is the linear operator given by $R_{\omega }\left(a\right):=a\;\omega$. Clearly, both $L_{\omega }$ and $R_{\omega }$ are positive, invertible linear operators on $A_{sa}$ because ω is a positive, invertible operator on H, and thus $A_{\omega }$ also is an invertible linear operator. It clearly holds
(127)$$a\;=\;A_{\omega }^{-1}\;\left(A_{\omega }\left(a\right)\right)\;=\;A_{\omega }^{-1}\;\left(\{\omega ,\;a\}\right),$$
and thus
(128)$$\begin{array}{cc} Z_{a}\left(\omega \right) & \;=\;Y_{A_{\omega }^{-1}\left(a\right)}\left(\omega \right)\\ Y_{a}\left(\omega \right) & \;=\;Z_{A_{\omega }\left(a\right)}\left(\omega \right) \end{array}$$
at every faithful positive linear functional $\omega \in P_{+}$. Furthermore, the gradient vector fields on the manifold $S_{+}\subset P_{+}$ of faithful states allow to identify a tangent vector at $\rho \in S_{+}$ with an element in $A_{sa}$ by means of
(129)$$Y_{a}\left(\rho \right)\;=\;\left\{\rho ,\;a\right\}-\mathrm{Tr}\left(\rho a\right)\;\rho .$$
However, since $S_{+}$ is the submanifold of $P_{+}$ determined by the inverse image of 1 with respect to the linear function $l_{I}^{+}$, and since a tangent vector at a point in $P_{+}$ may be identified with a self-adjoint element in $A_{sa}$, we obtain the identification of $T_{\rho }S_{+}$ with the linear subspace $A_{sa}^{0}\;\subset \;A_{sa}$ consisting of traceless elements mentioned before. Specifically, every constant vector field $Z_{a}$ in Equation (Equation 125) is tangent to $S_{+}$ whenever $a\in A_{sa}$ is such that Tr(a)=0. Consequently, there is a vector field $Z_{a}$ on $S_{+}$ to which $Z_{a}$ is $i_{1+}$-related.

Taking $\rho \in S_{+}$, the tangent vectors $a,b\in A_{sa}^{0}≅T_{\rho }S_{+}$, and $g\in \pi^{-1}\left(\rho \right)$, the Bures–Helstrom metric tensor $G_{BH}$ may be defined by (see [40] (Equation (Equation 3)))
(130)$${\left(G_{BH}\right)}_{\rho }\left(a,\;b\right)\;:=\;\inf\;\left\{\mathrm{Tr}\left(A^{†}\;B\right)\;\;|\;A,B\in \;T_{g}S_{1}\left(G\right)\;,\;T_{g}\pi \left(A\right)\;=\;a,\;\;\;T_{g}\pi \left(B\right)\;=\;b\;\right\}\;.$$
Then, it is possible to prove that (see [40,41,46]), given arbitrary tangent vectors $a,b\in A_{sa}^{0}≅T_{\rho }S_{+}$, the Bures–Helstrom metric tensor acts as
(131)$${\left(G_{BH}\right)}_{\rho }\left(a,\;b\right)\;=\;\mathrm{Tr}\left(a\;A_{\rho }^{-1}\left(b\right)\right)$$
where $A_{\rho }^{-1}$ is the inverse of $A_{\rho }$ (see Equation (Equation 126)), which is the invertible (because ρ is invertible) linear operator on B(H) given by $A_{\rho }\left(b\right)\;:=\frac{1}{2}\left(\rho \;b\;+\;b\;\rho \right)$. Note that the expression of $G_{BU}$ given in Equation (Equation 131) crucially depends on the identification of the tangent space $T_{\rho }S_{+}$ at $\rho \in S_{+}$ with the space of traceless, self-adjoint elements in B(H). That is, $G_{BH}$ is expressed in terms of the basis of vector fields $Z_{a}$ on $S_{+}$ that are $i_{1+}$-related with the constant vector fields $Z_{a}$ on $P_{+}$ defined in Equation (Equation 125), and, in this way, it becomes necessary to find the explicit form of the operator $A_{\rho }^{-1}$ at every ρ making the explicit computation of the action of the metric tensor not straightforward. On the other hand, the metric tensor $G_{1}$ in Equation (Equation 68) is “visualized” in terms of the gradient vector fields, and, with respect to these vector fields, its explicit expression is very easy to compute. We will now show that $G_{BH}$ and $G_{1}$ coincide because we have
(132)$$\begin{array}{cc} \left(G_{1}\left(Z_{a},\;Z_{b}\right)\right)\left(\rho \right) & \;=\;{\left(i_{1+}^{*}G\right)}_{\rho }\left(Z_{a}\left(\rho \right),\;Z_{b}\left(\rho \right)\right)\\ & \;=\;G_{\rho }\left(T_{\rho }i_{1+}\left(Z_{a}\left(\rho \right)\right),T_{\rho }i_{1+}\left(Z_{b}\left(\rho \right)\right)\right)\\ & \;=\;G_{\rho }\left(Z_{a}\left(\rho \right),\;Z_{b}\left(\rho \right)\right)\\ & \;=\;G_{\rho }\left(Y_{A_{\rho }^{-1}\left(a\right)}\left(\rho \right),\;Y_{A_{\rho }^{-1}\left(b\right)}\left(\rho \right)\right)\\ & \;=\;\mathrm{Tr}\left(\rho \;\{A_{\rho }^{-1}\left(a\right),A_{\rho }^{-1}\left(b\right)\}\right)\\ & \;=\;\mathrm{Tr}\left(\left\{\rho ,A_{\rho }^{-1}\left(a\right)\right\}\;A_{\rho }^{-1}\left(b\right)\right)\\ & \;=\;\mathrm{Tr}\left(a\;\;A_{\rho }^{-1}\left(b\right)\right)\\ & \;=\;{\left(G_{BH}\right)}_{\rho }\left(a,\;b\right)\;, \end{array}$$
where we used the first equality in Equation (Equation 128) in the fourth equality. From Equation (Equation 132), we conclude that $G_{1}=i_{1+}^{*}G$ is precisely the Bures–Helstrom metric tensor as claimed. This shows that with respect to the gradient vector fields, the explicit expression of the Bures–Helstrom metric is relatively easy to compute. The fact that we no longer have to find the explicit expression of the operator $A_{\rho }^{-1}$ at every ρ is due to the fact that we work with gradient vector fields, and we believe that this instance should be interpreted as an argument in favor of using the gradient vector because these vector fields better express the geometrical properties of the manifold of quantum states.

## 9. Concluding Remarks

In the context of quantum information theory, it is well-known that there is an infinite number of metric tensors on the manifold of faithful quantum states satisfying a property, which is the quantum analog of the monotonicity under Markov maps characterizing the Fisher–Rao metric tensor (up to a constant factor) in the classical case. The act of choosing which one of these metric tensors to use is thus an additional freedom that the quantum framework provides.

In this work, we presented a geometrical description of one of these quantum metric tensors, the so-called Bures–Helstrom metric tensor, which is rooted in the $C^{*}$-algebraic nature of the space of quantum states. Indeed, the theoretical framework of $C^{*}$-algebras allows to deal with classical probability distributions and quantum states “at the same time” because both of them can be realized as concrete realizations of the space of states on suitable $C^{*}$-algebras, and from this point of view, the algebraic structures on the algebra may be exploited to investigate the geometrical properties of the space of states. We focused on the finite-dimensional case, and we studied the geometrical structures associated with the symmetric part (Jordan product) of the associative product of the algebra, and the result is the definition of Riemannian metric tensors on submanifolds of states. In particular, we obtain that the Jordan product determines the Fisher–Rao metric tensor in the classical case, the Fubini–Study metric tensor in the case of pure quantum states, and the Bures–Helstrom metric tensor in the case of faithful quantum states, thus providing a theoretical framework in which all these seemingly different Riemannian metric tensors actually appear as different realizations of the “same” conceptual entity.

Finally, we want to mention that the geometrical picture outlined in this work heavily relies on the Jordan product of $A_{sa}$. As it is known, any associative algebra *A* over R gives rise to a Jordan algebra $A_{J}$ with Jordan product ⊙ given by $a⊙b:=\frac{1}{2}\left(a\cdot b+b\cdot a\right)$, where · is the associative product in *A* (see [51,52,54]). Consequently, it would be interesting to understand if and how it is possible to build a similar picture for the space of states of a Jordan algebra associated with an associative algebra that is not a $C^{*}$-algebra. In particular, “natural candidates” would be the many types of Geometric Algebras [73,74].

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
