# Peer review of "From the Jordan Product to Riemannian Geometries on Classical and Quantum States"

_entropy, 2020, doi:10.3390/e22060637_

Round 1

Reviewer 1 Report

fff

Author Response

See the pdf file

Reviewer 2 Report

The paper addresses the relation between algebra and geometry in the context
of C* algebras and the space of states. Within this approach the authors are
able to unify several Riemann structures when they particularize their
construction to different C^* algebras and/or different space of states.

I think the paper is rigourous it is well written and mostly self contained.
So it can be easily read.

I warn the authors of two mistakes or misprints in page 3.
First, it is not true that the anti-symmetric product [[ , ]]
is associative as the authors state, instead it satisfies Jacobi identity.
Second in formula (4) there is a "=0" missing.

A more general remark is that in several places in the paper
(at least in pages 8, 20, 21) the authors introduce the space
"of bounded linear operators on the finite-dimensional, complex Hilbert space".
No objection about the notation B(H) if they want to unify it with
the infinite dimensional case, but to mention "bounded" in this context
is, of course, redundant and actually misleading. I recommend the author to
suppress the reference to boundedness in the finite dimensional context
(which, besides, is the one they address).

Except for these minor amendments, I consider the paper adequate for publication.

Author Response

Concerning the mistake on $[[,]]$, we thank the referee for pointing it out. We provided to correct it immediately.

Concerning the missing $0$ in equation (4), we took care of it.

Concerning the use of the expression ``bounded linear operators'', we changed it into ``linear operators''.

Reviewer 3 Report

This paper studies the tensor fields induced on the manifold of positive linear functionals
by the commutator, respectively anti-commutator, of pairs of elements of a finite-dimensional C*-algebra.
The action of the group of invertible elements determines the geometry of the manifold.
The metric tensor is chosen to be the inverse of the tensor induced by the anti-commutators.
It is shown that this metric reduces to the canonical Fisher-Rao metric in the abelian case and to the Fubini-Study metric, respectively the Bures metric, in the quantum case.
This is an important result meriting publication.

Minor comments

p5, first line after (14): "starting at g(0)=g' is wrong: one has g(1)=g; g(0) is the identity.

p5, first line after (14): the symbol Gamma is normally used for the connection coefficients.
It is somewhat awkward to use it for a tangent vector.

p5, (16): For the reader it is much easier to start with (21) as a definition and then show that the Gammas are the fundamental vector fields of the action.

p5, line before (18): you could add that [,] denotes the Lie bracket: "...show that" --> "... show that the Lie bracket"
The whole sentence before (18) is rather cryptic.
The subsentence "..., since ...," is confusing rather than illuminating.

p6: I do not understand the paragraph containing (24,25). (24) is a definition. How can it mean something?
In addition, Y_ab coincides with Gamma_ab. Why two symbols for the same quantity?
Also openY_a and openX_a are redundant notations.

p7, l146: "Let us not fix the orbit O_1...". Is it 'not' or 'now'?
What do you intend to say?

On page 8 the tensor fields R and Lambda are defined. This surprises the reader because in the Introduction, at the top of page 2, these tensors are already defined with reference to the Literature.
I assume that you want to give an alternative definition and that you want to show that it reproduces the definition found in the Literature. If this is the case then you should say so.
Or you should repeat the definition as found in the Literature, with citations added,
and then make the connection with the present work.

p8, l168: " Moreover, that both tensors..." --> " Moreover, note that both tensors..."

p8, l184: "... every manifold O of positive linear functionals ..." every orbit?

p10, definition (52) and p12, display (69): the dependence of B_omega on b should be made explicit; for instance, the implicit dependence on b obscures the last line of (69). The former two lines of (69) depend implicitly on omega, the last line explicitly on rho. This distinction confuses the reader.

Author Response

See the pdf file
